# Completion of neural crest cell production and emigration is regulated by retinoic-acid-dependent inhibition of BMP signaling

Dina Rekler, Chaya Kalcheim*

Department of Medical Neurobiology, Institute of Medical Research Israel-Canada (IMRIC) and the Edmond and Lily Safra Center for Brain Sciences (ELSC), Hebrew University of Jerusalem-Hadassah Medical School, Jerusalem, Israel

**Abstract** Production and emigration of neural crest cells is a transient process followed by the emergence of the definitive roof plate. The mechanisms regulating the end of neural crest ontogeny are poorly understood. Whereas early crest development is stimulated by mesoderm-derived retinoic acid, we report that the end of the neural crest period is regulated by retinoic acid synthesized in the dorsal neural tube. Inhibition of retinoic acid signaling in the neural tube prevents the normal upregulation of BMP inhibitors in the nascent roof plate and prolongs the period of BMP responsiveness which otherwise ceases close to roof plate establishment. Consequently, neural crest production and emigration are extended well into the roof plate stage. In turn, extending the activity of neural crest-specific genes inhibits the onset of retinoic acid synthesis in roof plate suggesting a mutual repressive interaction between neural crest and roof plate traits. Although several roof plate-specific genes are normally expressed in the absence of retinoic acid signaling, roof plate and crest markers are co-expressed in single cells and this domain also contains dorsal interneurons. Hence, the cellular and molecular architecture of the roof plate is compromised. Collectively, our results demonstrate that neural tube-derived retinoic acid, via inhibition of BMP signaling, is an essential factor responsible for the end of neural crest generation and the proper segregation of dorsal neural lineages.

*For correspondence:
kalcheim@cc.huji.ac.il

Competing interest: The authors declare that no competing interests exist.

## Editor's evaluation

This manuscript examines how retinoic acid signaling controls the timing of neural crest production in avian embryos. The authors propose that local production of retinoic acid signaling activates the expression of BMP inhibitors in the dorsal neural tube. Disruption of BMP signaling results in the termination of neural crest migration and the establishment of the definite roof plate. In the absence of RA signaling cells in the dorsal neural tube continue to express neural crest markers well into the roof plate stage, and neural crest delamination is prolonged, but interestingly this does not prevent roof plate-mediated specification of dorsal interneurons.

## Introduction

The generation of cell type diversity during normal development takes place in a stereotypic sequence and a spatially regulated manner (*Holguera and Desplan, 2018*; *Kohwi and Doe, 2013*). An example of such a precision is the development of the central and peripheral subdivisions of the nervous system (CNS and PNS), that arise from a common domain in the dorsal neural tube (NT). This domain

**eLife digest** The division between the central nervous system – formed by the brain and spinal cord – and the peripheral nervous system – which consists of the neurons that sense and relay information to and from the body – takes place early during embryonic development. Initially, the nervous system consists of a tube of cells called the neural tube. From the top region of this tube, some cells change their shape, exit the tube and migrate to different places in the developing body. These cells are called the 'neural crest', and they form many different structures, including the peripheral nervous system.

Neural crest cells keep leaving the neural tube for a period of time, but after that, the neural tube stops producing them. At this point, the region of the neural tube that had been producing neural crest cells becomes the 'roof plate' of the central nervous system, a structure that is essential for the development of specific groups of neurons in the brain and spinal cord.

In bird embryos, a protein called bone morphogenetic protein (BMP) is essential for neural crest production because it triggers the migration of these cells away from the neural tube. Before the roof plate is formed, the activity of BMP is blocked by proteins known as BMP inhibitors, which stop more cells from leaving the neural tube. Around the time when neural crest formation stops, another molecule called retinoic acid begins to be synthesized in the top region of the neural tube. Rekler and Kalcheim asked whether retinoic acid is involved in the transition from neural crest to roof plate.

To test this hypothesis, Rekler and Kalcheim blocked the activity of retinoic acid in the neural tube of quail embryos at the time when they should stop producing neural crest cells. This resulted in embryos in which the neural tube keeps producing neural crest cells after the roof plate has formed. In these embryos, individual cells in the resulting 'roof plate' produced both proteins that are normally only found in neural crest cells, and proteins typically exclusive to the roof plate. This suggests that, in the absence of retinoic acid activity, the segregation of neural crest identity from roof plate identity is compromised.

Rekler and Kalcheim also found that, in the embryos where retinoic acid activity had been blocked, the cells in the area where the roof plate should be produced virtually no BMP inhibitors, and exhibited extended BMP activity. This allowed neural crest cells to continue forming and migrating away from the neural tube well after the period when they would stop in a normal embryo. These results indicate that retinoic acid stops the production of neural crest cells by repressing BMP activity in the roof plate of the neural tube.

Rekler and Kalcheim's experiments shed light on the mechanisms that allow the central and peripheral nervous systems to become segregated. This could increase our understanding of the origin of several neurodevelopmental disorders, potentially providing insights into their treatment or prevention. Additionally, the process of neural crest production and exit from the neural tube is highly similar to the process of metastasis in many invasive cancers. Thus, by understanding how the production of neural crest cells is terminated, it may be possible to learn how to prevent malignant cancer cells from spreading through the body.

sequentially generates neural crest cells (NC), progenitors of the peripheral nervous system (PNS) (*Bronner, 2012*; *Le Douarin and Kalcheim, 1999*), followed by the definitive roof plate (RP) of the CNS, which is flanked ventrally by dorsal interneuron populations (*Andrews et al., 2019*; *Chizhikov and Millen, 2005*; *Helms and Johnson, 2003*; *Kalcheim and Kumar, 2017*; *Krispin et al., 2010a*; *Nitzan et al., 2016*).

At trunk levels of the avian axis, NC progenitors sequentially emigrate from the NT for a period of about two days while actively proliferating before and after delamination and largely synchronizing to the S-phase of the cell cycle during the emigration event (*Burstyn-Cohen and Kalcheim, 2002*). Lineage-tracing experiments showed that RP progenitors originate ventral to the cohort of premigratory NC (*Krispin et al., 2010b*). As a result of continuous cell emigration, RP precursors undergo a ventro-dorsal shift and, upon completion of NC departure, they reach the dorsal midline of the NT and form the definitive RP (*Krispin et al., 2010b*). RP progenitors gradually exit the cell cycle to become post-mitotic (*Kahane and Kalcheim, 1998*), and function primarily as a dorsal organizer, providing a gradient of Bone Morphogenetic Protein (BMP) and Wnt activity which is crucial for development

of dorsal interneurons (*Chizhikov and Millen, 2005*; *Le Dréau and Martí, 2013*; *Muroyama et al., 2002*; *Zechner et al., 2007*) and for proliferation of ependymal cells at later stages (*Shinozuka et al., 2019*; *Xing et al., 2018*).

Many aspects of early NC ontogeny were extensively studied, from induction to specification, epithelial to mesenchymal transition (EMT) and emigration. Yet, virtually nothing is known about the mechanisms underlying the end of NC production and transition to definitive RP (*Rekler and Kalcheim, 2021*).

BMP has been found to regulate several aspects of NC development. In the early neurula, BMP is derived from the non-neural ectoderm and participates in NC induction at the neural plate border (*Endo et al., 2002*). Upon closure of the NT, cells of the dorsal domain synthesize and secrete BMP, which acts in a graded rostro-caudal fashion to activate Wnt signaling and a network of downstream genes that altogether promote NC EMT and emigration (Reviewed in *Kalcheim, 2018*; *Theveneau and Mayor, 2012a*).

A first step toward understanding the segregation between NC and RP, was the finding that approaching the end of the NC production in the avian embryo, the dorsal NT becomes insensitive to BMP, even though this factor continues to be produced by nascent RP cells. This suggested that local downregulation of BMP signaling is necessary for the end of NC stage, a conclusion further illustrated by a premature cessation of NC proliferation and delamination upon early inhibition of BMP signaling (*Nitzan et al., 2016*). These events were mediated by the transcription factor HES1/Hairy1, as early misexpression of *hes1/hairy1* in the dorsal NT led to a downregulation of the *Alk3* BMP receptor and of pSmad activity (*Nitzan et al., 2016*).

However, the factors that orchestrate these BMP-dependent processes, leading to cessation of NC production, segregation of NC and RP lineages, and formation of the definitive RP, remain unknown. Along this line, whether cessation of NC production and RP development are regulated by the same or by independent mechanisms is yet to be elucidated.

A recent RNA-seq analysis revealed transcriptional heterogeneity between premigratory NC and definitive RP cells, uncovering a set of genes expressed specifically at the onset of the RP stage (*Ofek et al., 2021*). One of these genes is *Raldh2/Aldh1a2*, that encodes an enzyme responsible for the biosynthesis of the morphogen retinoic acid (RA) (*Haselbeck et al., 1999*), central to multiple neuro-developmental processes (*Diez del Corral and Morales, 2014*; *Lara-Ramírez et al., 2013*; *Wilson et al., 2004*). RA derived from the *Raldh2*-expressing somitic mesoderm has been described as a key factor in NT development, including control of motoneuron specification (*Novitch et al., 2003*; *Wilson et al., 2004*) and initiation but not maintenance of NC EMT and emigration (*Martínez-Morales et al., 2011*). Although expression of *Raldh2* in the dorsal NT was previously reported (*Berggren et al., 1999*; *Blentic et al., 2003*), its precise temporal dynamics vis-à-vis NC/RP phases and function/s have not been characterized. Hence, the discovery of local RA production in the dorsal NT only by the end of the NC stage prompted us to hypothesize a role for RA in events concerning the late NC.

In this study, we unravel a fundamental role of dorsal NT-derived RA in controlling the end of NC production. Using two separate tools to inhibit RA signaling in the quail embryo, we demonstrate that RA is responsible for downregulation of BMP signaling in the dorsal NT. In the absence of RA signaling, cells in the dorsal NT retain expression of NC markers well into the RP stage which also displays additional NC traits such as a high mitotic rate and impaired epithelial properties. Furthermore, we show that lack of RA signaling prolongs cell delamination into the RP stage. The latter effect is abrogated by concomitant inhibition of BMP signaling, suggesting that the RA-dependent extension of NC EMT is also mediated by BMP. Whereas RA signaling represses expression of NC genes such as *foxd3, snai2,* and *sox9*, extending the activity of the latter beyond the NC stage inhibits transcription of *Raldh2* in the RP, suggesting that NC and RP traits stand in a mutual cross-repressive interaction. Remarkably, maintenance of NC traits in the dorsal NT does not prevent the advent of RP traits, or RP-dependent specification of dorsal interneurons. Rather, in the absence of RA activity, the spatial and temporal segregation of dorsal lineages, as well as the structural integrity of this signaling center, are compromised.

## Results

### RA is both produced and active in the nascent RP

The spatio-temporal dynamics of RA, including synthesis, degradation, and activity, was examined in the trunk of quail embryos from stages corresponding to NC production until formation of the nascent RP. RA is produced from retinol via two consecutive reactions: first, retinol is converted to retinaldehyde by RDH enzymes, then it is oxidized to RA by RALDH/ALDH1; additionally, CYP1B1 catalyzes both reactions (*Figure 1A* and *Chambers et al., 2007*; *Kam et al., 2012*; *Reijntjes et al., 2005*; *Rhinn and Dollé, 2012*). Of the three known RALDH-encoding genes, only *Raldh2* is expressed in the avian trunk at these developmental stages (*Blentic et al., 2003*). At an early NC stage, *Raldh2* was expressed in the intermediate and paraxial somitic mesoderm but not in the neuroepithelium (*Figure 1B*). At an advanced NC stage, *Raldh2* continued to be transcribed in subdomains of the somite-derived sclerotome and dermomyotome (*Figure 1C*) and was weakly detected in the dorsal NT (*Figure 1C*, arrowhead). Subsequently, *Raldh2* was strongly expressed in the nascent RP itself (*Figure 1D*, arrowhead, *Figure 1—figure supplement 1A*). In contrast, no expression of *Cyp1B1* was apparent in the NT at any stage, although robust transcription was evident in the paraxial mesoderm throughout development (*Figure 1E–G*).

RA is degraded by CYP26 enzymes of the cytochrome P450 family. *Cyp26A1* is the only member of the family to be expressed in the avian trunk at early stages of nervous system development (*Blentic et al., 2003*). At an early NC stage, *Cyp26A1* was absent from the NT and adjacent mesoderm, it was later detected in a salt and pepper pattern in dorsal interneuron progenitors, but was excluded from the dorsal midline (*Figure 1H–I*). As the definitive RP formed, it was also occasionally detected in RP cells (*Figure 1J*).

CRABP1 is a carrier protein of RA, which participates in regulation of RA function and/or metabolism (*Maden et al., 1989*; *Venepally et al., 1996*). At the early NC stage, there was no prominent expression of *CRABP1* in either the NT or paraxial mesoderm (*Figure 1K*); at an advanced NC stage it appeared in the basal domain of the NT, corresponding to the positions adopted by differentiating neurons, yet absent from the dorsal and ventral midline regions (*Figure 1L*). At the RP stage, *CRABP1* was additionally expressed in two stripes at the lateral domain of the RP (*Figure 1M*, *Figure 1—figure supplement 1A*, and see *Ofek et al., 2021*).

Next, we examined expression of RA receptors. RA acts by binding to a heterodimer of nuclear receptors, RAR and RXR, each of which is encoded by three genetic variants. Expression of all RA receptors was predominantly ubiquitous in both early NC (*Diez del Corral et al., 2003*) and RP stages with at least one variant of each subunit expressed in the dorsal NT at the nascent RP stage (RARα, RARβ and RXRγ, but not RARγ or RXRα) (*Figure 1—figure supplement 2*). This suggests that RA acts during the transition into the nascent RP where a tight regulation of local RA levels may take place.

To directly examine the activity of RA in the dorsal NT, we electroporated a destabilized GFP plasmid driven by a retinoic acid response element (RARE-d2EGFP) at E2.5 and analyzed GFP fluorescence 12 and 48 hours later, at NC and RP stages, respectively. Whereas control RFP was apparent throughout the dorso-ventral extent of the NTs, RARE activity was primarily present within the dorsal half of the NT at both stages (*Figure 1N–O'*).

Together, these data show that whereas NC cells respond to mesoderm-derived RA (see Introduction for refs.), but do not synthesize it, the emerging RP both synthesizes and responds to this factor. Hence during the transition from NC to definitive RP, the latter becomes a local source of RA. This prompted us to hypothesize that locally produced RA plays a role in either the end of NC production and emigration, in the ensuing formation of the RP, or in both processes.

### RA is responsible for the loss of sensitivity of dorsal NT progenitors to BMP during the transition from NC to RP

In previous studies, we showed that BMP signaling is essential for NC EMT and emigration (*Sela-Donenfeld and Kalcheim, 1999*). Subsequently, when approaching the RP stage, although BMP is still produced by the nascent RP, the latter becomes refractory to it, likely via downregulation of BMP receptor 1 A (*Alk3*), and upregulation of BMP inhibitory genes (*Figure 1—figure supplement 1B*; *Nitzan et al., 2016*; *Ofek et al., 2021*). The onset of *Raldh2* synthesis close to RP development prompted us to hypothesize that RA, locally produced in the dorsal NT, regulates this dynamic cellular behavior vis-a-vis BMP. To this end, we implemented two independent means to inhibit RA activity in

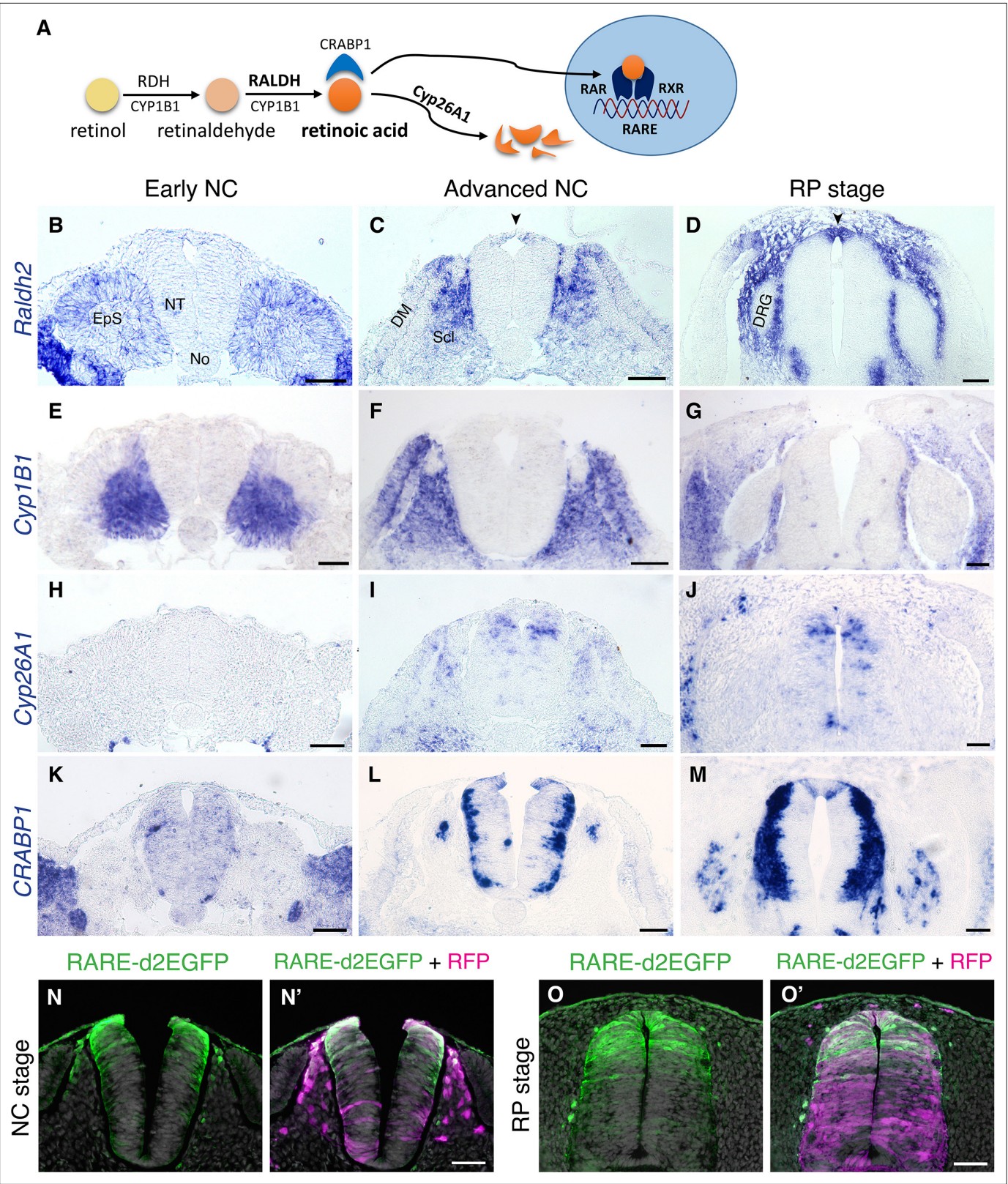

**Figure 1.** RA is both produced and active in the nascent RP. (**A**) Schematic illustration of the metabolic and signaling pathways of RA. (**B–M**) Expression patterns of genes participating in RA synthesis and metabolism, both at early and advanced NC stages [E2 (27 ss) and E3 (35 ss), respectively], and at the RP stage (**E4**). ISH for *Raldh2/Aldh1a2* (**B–D**), *Cyp1B1* (**E–G**), *Cyp26A1* (**H–J**), and *CRABP1* (**K–M**). Images were taken at the level of somites 25–27. (**N-O'**) RA activity in the NT at NC and RP stages, indicated by the presence of destabilized GFP (d2EGFP) driven under the control of a RARE element.

*Figure 1 continued on next page*

*Figure 1 continued*

Embryos were electroporated at E2.5 (28 ss) along with control RFP to monitor efficiency of transfection, and analyzed 12 hr (N-N', at 35 ss) or 48 hr (O-O', at E4) later. RARE activity was evident in the dorsal NT of NC-stage embryos (N = 8/8), and RP-stage embryos (N = 7/7). Images were taken at the level of somites 25–27. Abbreviations, EpS, epithelial somite, No, notochord, NT, neural tube, DM, dermomyotome, Scl, sclerotome, DRG, dorsal root ganglion. Scale bar, 50 µm.

The online version of this article includes the following figure supplement(s) for figure 1:

**Figure supplement 1.** Expression patterns of selected genes from a comparative RNA-seq analysis between NC and RP stages.

**Figure supplement 2.** Expression of RA receptors in the dorsal NT at the RP stage.

vivo: (1) RARα403 - a truncated human RA receptor that lacks the transactivation domain and therefore acts in a dominant-negative fashion (*Gupta and Sen, 2015*), and (2) Cyp26A1 which encodes a RA-degrading enzyme.

First, the efficacy of these constructs was tested by co-electroporation with RARE-RFP. RFP intensity was significantly reduced by 79% and 88% in Cyp26A1- and RARα403-electroporated embryos, respectively, when compared to GFP-only controls. Similarly, RARα403 almost completely abolished the RARE-AP signal observed in controls (*Figure 2—figure supplement 1*). Thus, both RARα403 and Cyp26A1 can be effectively used to examine the consequences of RA signaling inhibition.

Next, we investigated the effect of RA on the dynamics of BMP signaling. First, we monitored pSmad 1/5/9 expression, a readout of BMP activity, at the NC stage. Control GFP was delivered to embryos aged 27 somite pairs followed by fixation 12 hr later. pSmad was strongly expressed in the dorsal NT (*Figure 2A and A'*), corroborating previous findings (*Nitzan et al., 2016*). In parallel, NTs were electroporated at E2.5 (27 somite pairs at the level of epithelial somites, NC stage) with either RARα403 or Cyp26A1 and analyzed 2 days later (E4, RP stage). As expected (*Nitzan et al., 2016*), the RP of control embryos was already devoid of BMP activity at this stage (*Figure 2B and B'*, arrows). In contrast, the RP domain of embryos treated with either RARα403 or Cyp26A1 was pSmad-positive, indicating a failure to normally downregulate BMP signaling (*Figure 2C–D'*, arrowheads). The dI1 progenitor population of dorsal interneurons, located ventral to the RP, was also pSmad-positive, consistent with its dependency on BMP for proper development (*Duval et al., 2019*; *Lee et al., 1998*; *Tozer et al., 2013*; *Wine-Lee et al., 2004*). Quantification of pSmad intensity confirmed a significant increase in Cyp26A1 and RARα403-treated embryos compared to controls. This was apparent even if measurement of pSmad intensity comprised the dorsal NT from its midline to the ventral border of pSmad$^+$ staining, that included dI1 progenitors (*Figure 2E*). These results show that inhibition of RA signaling extends BMP activity into the RP stage by preventing its normal downregulation during the transition from NC to RP.

We next examined whether the observed RA-dependent downregulation of BMP activity is mediated by a corresponding upregulation of BMP inhibitors at the RP stage. In this context, Hairy1 was previously reported to act as a specific RP-derived BMP inhibitor thus participating in the end of NC production (*Nitzan et al., 2016*). Furthermore *hairy1/qHes4*, and the known BMP inhibitors *BAMBI* (*Brazil et al., 2015*) and *Grem1* (*Sun et al., 2006*), were unraveled as specific RP markers in a recent RNA-seq analysis comparing dorsal NT cells at both premigratory NC and RP stages (*Figure 1—figure supplement 1B*; *Ofek et al., 2021*). Embryos electroporated with either Cyp26A1 or RARα403 at E2.5 exhibited a substantially decreased intensity of *BAMBI* mRNA staining in the RP relative to controls (*Figure 2F–I*). Similarly, RARα403 significantly affected the upregulation of both *hairy1* and *Grem1* in RP when compared to control embryos (*Figure 2J–O*). These results show that the downregulation of BMP signaling by RA is accounted for, at least partially, by upregulation of BMP inhibitory genes.

In a previous study, we reported that BMP-dependent emigration of NC cells is mediated by Wnt activity (*Burstyn-Cohen et al., 2004*). A dynamic behavior of Wnt pathway genes between NC and RP stages was also evident in a RNA-seq analysis (*Figure 1—figure supplement 1C*). This raised the possibility that, in the absence of RA, extended BMP signaling at the RP stage would be associated with enhanced Wnt activity. To this end, we assessed Wnt signaling in NC and RP by electroporating a destabilized Wnt reporter, 12XTopFlashd2GFP, together with control RFP. Whereas significant activity of the Wnt pathway was observed in the early dorsal NT and emigrating NC cells, a substantial decrease in 12XTopFlashd2GFP expression was apparent in the normal RP (*Figure 2—figure supplement 2A-B'*, D). In contrast, a significantly enhanced expression was measured in the RP of

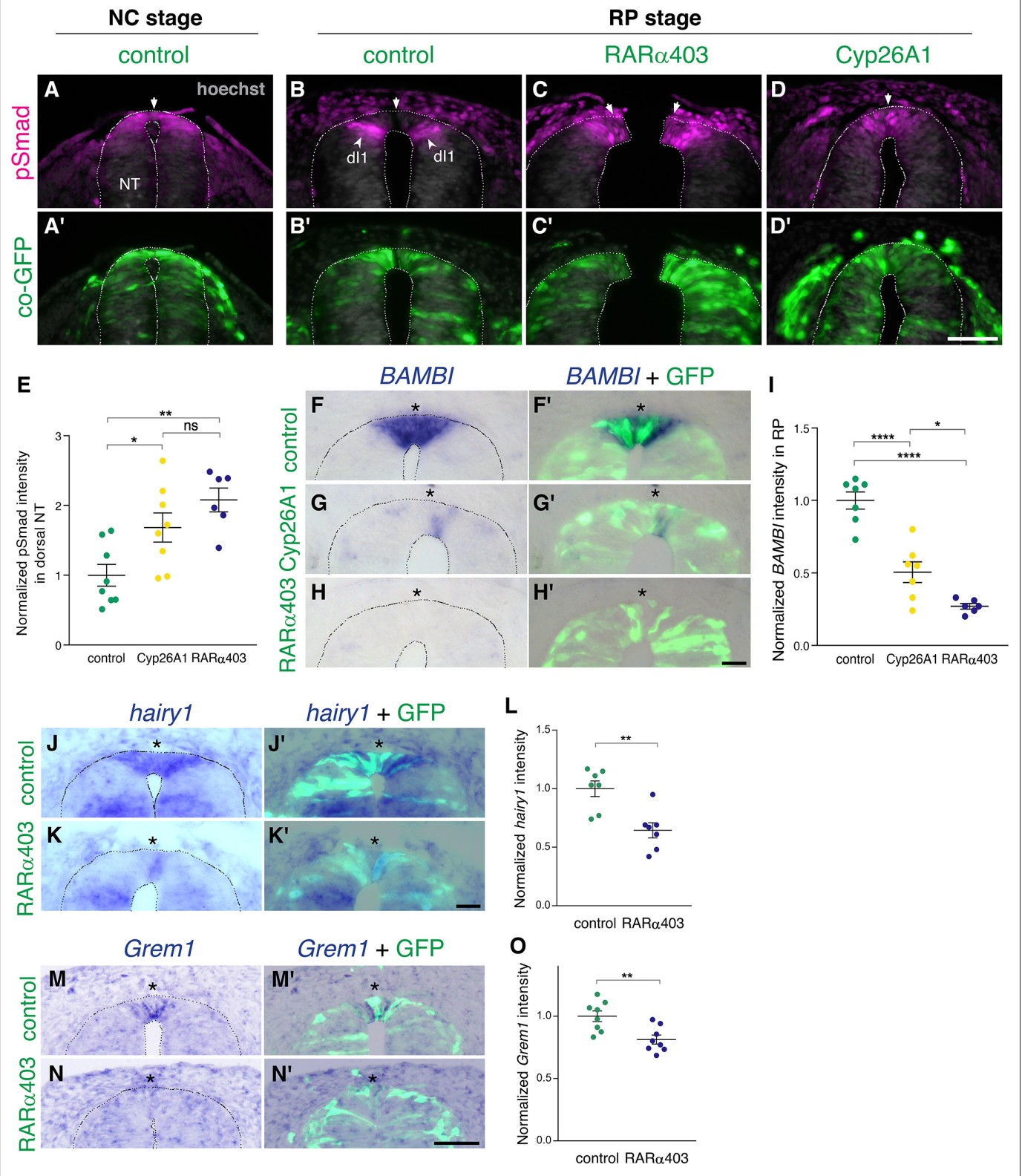

**Figure 2.** RA is responsible for the loss of sensitivity of dorsal NT progenitors to BMP during the transition from NC to RP. (**A-D'**) Immunostaining for pSmad1/5/9. Embryos at E2.5 (27-8ss) were co-electroporated with RARα403, Cyp26A1 or with a control PCAGG vector, along with GFP, and analyzed at 35ss (control only, NC stage) and at E4. Note the presence of BMP activity in premigratory NC (**A**) and its absence in the RP of control embryos (N = 8, **B-B'**). Whereas only ventrally located dorsal interneurons are positive in control embryos, the RP domain itself is positive in Cyp26A1 (N = 8) and

*Figure 2 continued on next page*

*Figure 2 continued*

RARα403-electroporated embryos (N = 7, **C-C',D-D'**). (**E**) Quantification of pSmad staining intensity (measured area includes dorsal interneurons). Imaging and quantification were performed at somite levels 24–26. Four-to-21 sections per embryo were analyzed. *p < 0.05, **p < 0.005, one-way ANOVA with post-hoc Tukey's tests. (**F–O**) In situ hybridization of embryos electroporated with Cyp26a1 or RARα403 at E2.5 (27 ss) and analyzed at E4. Asterisks mark the RP domain. (**F–I**) ISH for *BAMBI*, showing downregulation in the RP of Cyp26A1- and RARα403-treated embryos. (**I**) For quantification of the intensity of *BAMBI*, 6-to-21 sections per embryo were analyzed at somite levels 24–26. N = 7,7 and 6 embryos for control, Cyp26A1 and RARα403, respectively. *p < 0.05, ****p < 0.0001, one-way ANOVA with post-hoc Tukey's tests. (**J–L**) ISH for *hairy1*, showing downregulation in the RP of RARα403-treated embryos. (**L**) For quantification of the intensity of *hairy1*, 7–18 sections per embryo were analyzed at somite levels 24–26. N = 7 embryos for each group. **p < 0.005, Student's unpaired t-test. (**M–O**) ISH for *Grem1*, showing downregulation in the RP of RARα403-treated embryos. For quantification of the intensity of *Grem1*, 11–27 sections per embryo were analyzed at somite levels 24–26. N = 8 embryos for each group. **p < 0.005 via Student's unpaired t-test. Abbreviations, dl1, dorsal interneurons 1, RP, roof plate. Scale bar, 50 μm.

The online version of this article includes the following source data and figure supplement(s) for figure 2:

**Source data 1.** RA is responsible for the loss of sensitivity of dorsal NT progenitors to BMP during the transition from NC to RP.

**Figure supplement 1.** Electroporation of Cyp26A1 or RARα403 into the NT effectively downregulates RA activity.

**Figure supplement 1—source data 1.** Electroporation of Cyp26A1 or RARα403 into the NT effectively downregulates RA activity.

**Figure supplement 2.** RA promotes the downregulation of Wnt signaling in the nascent RP.

**Figure supplement 2—source data 1.** RA promotes the downregulation of Wnt signaling in the nascent RP.

RARα403-transfected NTs (*Figure 2—figure supplement 2C-D*). These results show that inhibition of RA signaling extends Wnt activity into the RP stage by preventing its normal downregulation during the transition from NC to RP. Hence, RA is necessary for the downregulation of BMP and of downstream Wnt signaling towards ending NC production and/or emigration.

## RA downregulates NC markers during the transition to definitive RP

Our results prompted us to investigate whether the observed persistence of BMP and Wnt activities into the RP stage is associated with a retention of NC identity.

*Foxd3*, *snai2*, and *sox9* are known premigratory markers of trunk NC cells (*Nieto et al., 1994*; *Sela-Donenfeld and Kalcheim, 1999*), and are downregulated in the dorsal NT upon emigration of neural progenitors, being absent from prospective melanoblasts and RP (*Krispin et al., 2010b*; *Nitzan et al., 2013*; *Ofek et al., 2021*). Expression of *foxd3*, *snai2*, and *sox9* transcripts was examined at the RP stage in embryos electroporated with RARα403. Strikingly, whereas no mRNA expression was detected in controls, all three genes persisted in the dorsal NT of treated embryos (*Figure 3A–F'*, arrowheads). Moreover, using immunohistochemistry to assess Sox9 protein, we confirmed its maintenance in the dorsal NT at the RP stage both in Cyp26A1- and RARα403-electroporated embryos (*Figure 3G–J*).

These findings suggest that endogenous RA signaling underlies the loss of NC identity towards the end of the NC period.

## RA signaling promotes cell cycle exit and epithelial traits in dorsal NT cells

NC cells proliferate extensively while residing in the NT and also during migration. Upon the end of NC production, dorsal NT progenitors gradually exit from the cell cycle to become post-mitotic RP cells (*Kahane and Kalcheim, 1998*; *Nitzan et al., 2016*). We asked whether RA inhibition affects cell cycle withdrawal of nascent RP cells. To this end, we immunostained embryos at E4 (RP stage) for the mitotic marker phosphohistone H3 (pH3), following electroporation of RARα403 or Cyp26A1 at E2.5 (NC stage). Remarkably, a significantly higher number of mitotic nuclei was observed in the RP of embryos treated with either Cyp26A1 or RARα403 compared to controls which exhibited a low extent of mitosis (*Figure 4A–D*). These results suggest that endogenous RA inhibits cell proliferation, consistent with previous data obtained in forebrain and optic tectum of avian embryos (*Gupta and Sen, 2015*; *Kukreja et al., 2020*).

Furthermore, characteristic of post-mitotic, differentiating progenitors is the localization of nuclei to the basal domain of the neuroepithelium, as observed in control embryos at the RP stage (*Figure 4E*). In contrast, in RARα403-electroporated embryos, nuclei were spread throughout its apico-basal extent. Quantification showed that their localization to the apical half of the epithelium significantly

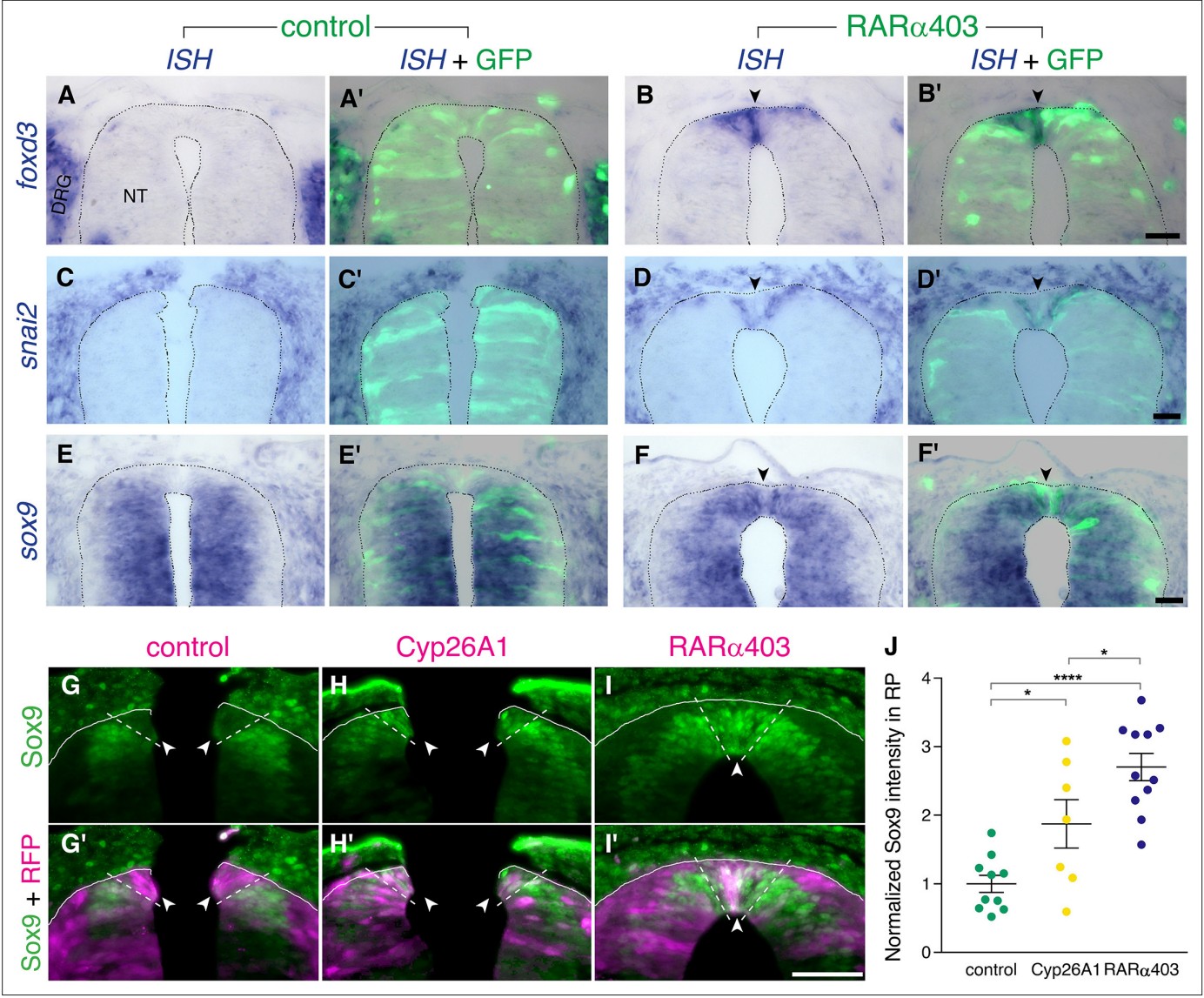

**Figure 3.** RA downregulates NC markers during the transition to definitive RP. (**A-F′**) ISH for neural crest markers on embryos electroporated with GFP along with control PCAGG or RARα403 at E2.5 (27ss) and analyzed at E4. Arrowheads point to the RP in RARα403-treated embryos. (**A-B′**) ISH for *foxd3*, showing no expression in the RP of control embryos (N = 7) compared to strong expression in the RP of all RARα403-treated embryos (N = 7). (**C-D′**) ISH for *snai2*, showing no expression in the RP of control embryos (N = 11) and clear expression in the RP of all RARα403-treated embryos (N = 11). (**E–F**) ISH for *sox9*, showing no expression in the RP of control embryos (N = 9) and strong expression in the RP of all RARα403-treated embryos (N = 9). Note as well expression of *sox9* mRNA in more ventral regions of the NT and in adjacent sclerotome. Imaging and analysis were performed at somite levels 25–27. (**G–J**) Immunostaining for Sox9 in control and experimental embryos, showing the presence of Sox9 protein in the RP of Cyp26A1- and RARα403-treated embryos (arrowheads), compared to absence of Sox9 in the RP of control embryos. RFP was co-electroporated in both groups. Dashed lines delineate the RP domain. (**J**) Measurement of Sox9 staining intensity. For quantification, 15 sections per embryo were analyzed. Imaging and analysis were performed at somite levels 25–27. N = 10, 7, and 11 embryos for control, Cyp26A1 and RARα403 groups, respectively. *p < 0.05, ****p < 0.0001, one-way ANOVA with post-hoc Tukey's tests. Abbreviations, NT, neural tube, DRG, dorsal root ganglion. Scale bar, 50 µm.

The online version of this article includes the following source data for figure 3:

**Source data 1.** RA downregulates NC markers during the transition to definitive RP.

exceeds that seen in control conditions (*Figure 4E–G*). This is consistent with the higher extent of cell proliferation monitored in the absence of local RA signaling, reflecting the retention of interkinetic nuclear migration with presence of mitotic nuclei in the ventricular zone, and/or a disorganization of the nascent RP under experimental conditions.

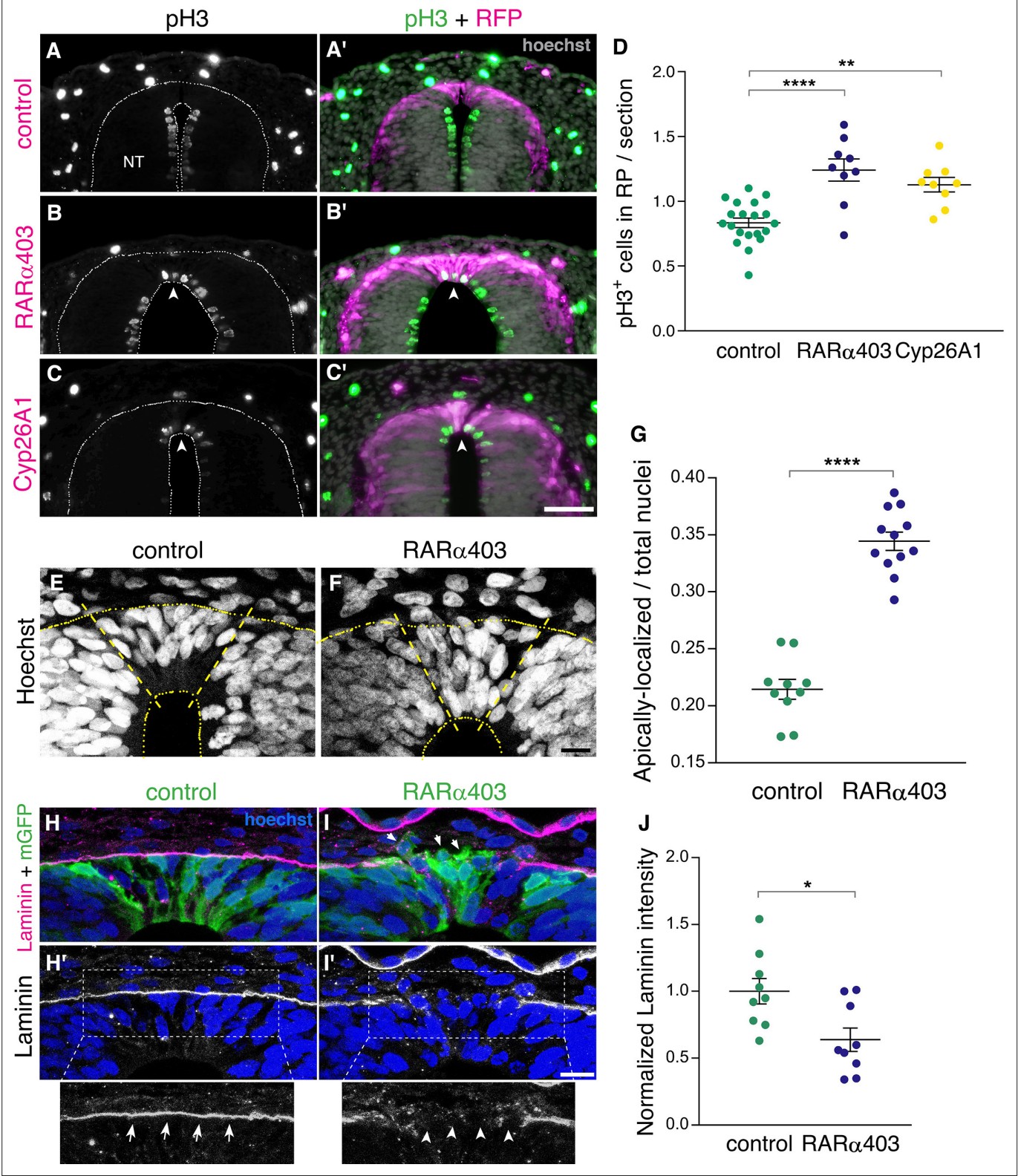

**Figure 4.** RA signaling promotes cell cycle exit and epithelial traits in dorsal NT cells. (**A–D**) Immunostaining for phospho-Histone H3 of embryos electroporated at E2.5 (27 ss) with either control PCAGG, RARα403 or Cyp26A1, and analyzed at E4. Arrowheads in B-C' point to pH3+ cells in the RP domain of RARα403- and Cyp26A1-electroporated embryos. The quantification of pH3+ cells in the RP is represented in (**D**). 120 sections were analyzed per embryo. Imaging and analysis were performed at somite levels 24–26. N = 20, 9 and 9 embryos for control, RARα403 and Cyp26A1, respectively. **p < 0.005, ****p < 0.0001, one-way ANOVA with post-hoc Tukey's tests. (**E–G**) Apico-basal localization of Hoechst-stained nuclei in

*Figure 4 continued on next page*

*Figure 4 continued*

embryos electroporated as in A-D. Whereas in control embryos most nuclei are restricted to the basal domain of the RP, in the RP of RARα403-treated embryos, nuclei are apparent across the entire thickness of the tissue. Confocal images were z-stacked and nuclei in the apical and basal halves of the RP domain were counted. The proportion of apically-localized nuclei is represented in (**G**). Thirteen-to-17 sections were analyzed per embryo. Imaging and analysis were performed at somite levels 25–27. N = 10 and 12 embryos for control and RARα403 groups, respectively, ****$p < 0.0001$, exact p-value Mann-Whitney test. (**H–J**) Immunostaining for Laminin of embryos electroporated as in A-D. Arrows and arrowheads in insets of H' and I' indicate the presence of either a continuous or a discontinuous RP basal lamina, respectively. Arrows in I point to delaminating cells. (**J**) Quantification of Laminin intensity along the basal aspect of the RP. Six-to-18 sections were analyzed per embryo. Imaging and analysis were performed at somite levels 25–27. N = 9 embryos both for control and RARα403 treatments. *$p < 0.05$, Student's unpaired t-test. Abbreviations, NT, neural tube. Scale bar, (**A–C**), 50 μm, (**E–F, H–I**), 10 μm.

The online version of this article includes the following source data for figure 4:

**Source data 1.** RA signaling promotes cell cycle exit and epithelial traits in dorsal NT cells.

Another critical process that occurs following the end of NC emigration is the restoration of epithelial cell traits in the nascent RP (*Nitzan et al., 2016*). One of these traits is the deposition of a continuous and prominent laminin-containing basal lamina covering the normal RP (*Figure 4H and H'*). This basal lamina is incomplete at the NC stage due to continuous cell delamination. Inhibition of RA signaling resulted in a weaker and discontinuous pattern of laminin immunostaining dorsal to the RP (*Figure 4I–J*), indicative of the maintenance of NC characteristics at the later RP stage. Consistently, upon electroporation with membrane-tethered GFP, it was apparent in controls that cells extended along the apico-basal domain of the RP (*Figure 4H*). In contrast, in the absence of RA activity, many cells were devoid of epithelial integrity, including labeled cells detaching from the epithelium (*Figure 4I*, arrows). Combined, these results show that inhibition of RA signaling in the dorsal NT prevents cells from exiting the cell cycle and from adopting cellular traits that characterize the normal ontogeny of a RP.

## RA is responsible for the end of NC delamination

As described above, inhibiting RA activity in the dorsal NT extends the period of BMP activity and preserves NC properties such as premigratory genes and cell proliferation into the RP stage. We therefore asked whether the most discernible difference between NC cells and definitive RP cells, the process of EMT and cell delamination, is prolonged as well by manipulation of RA signaling.

To this end, we implemented a double electroporation protocol: first, we electroporated RARα403 along with RFP only to one hemi-NT at E2 and at E3.5 both sides were labeled with GFP-DNA (*Figure 5A*). By E4, RFP-positive NC cells exited the NTs under both control and treated conditions primarily toward the side ipsilateral to the transfection and also to some extent towards the contralateral side, as previously described (*Burstyn-Cohen and Kalcheim, 2002*; *George et al., 2007*). However, the number of emigrated, late GFP-labeled cells was significantly higher on the RARα403-electroporated side when compared to the control contralateral side and also to both sides of control embryos (*Figure 5B–D*, arrowheads in C'). These data show that RA signaling promotes the end of NC cell delamination.

Even though we observed that loss of RA activity stimulates sustained pSmad expression (*Figure 2B–D*) and abrogates transcription of BMP inhibitors (*Figure 2F–O*) along with extended NC emigration (*Figure 5A–D*), it is possible that this late EMT elicited by loss of RA is BMP-independent. If that were true, then inhibition of BMP with Smad6 (*Nitzan et al., 2016*), should be without effect on cell emigration induced by RARα403 at the RP stage.

To test this hypothesis, embryos aged E2 were bilaterally electroporated (*Figure 5A*, bottom) with either control GFP, RARα403 to induce late NC delamination (*Figure 5B–D*), with Smad6, or with both RARα403 and Smad6. At E3.5, control GFP was bilaterally delivered to all embryos and the outcome was analyzed at E4 (*Figure 5A*). In all cases, RFP-labeled cells that indicate the efficiency of early transfections, successfully delaminated from the NT. In contrast, under control conditions, few or no GFP-labeled cells delaminated from the NT (*Figure 5E–E'1*), consistent with the second electroporation being done following the end of NC emigration. As also shown above, in RARα403-treated embryos, a substantial number of GFP-labeled cells had emigrated (*Figure 5F–F'1* and see also D). As expected from late electroporations, treatment with Smad6 alone resembled the control situation with few delaminated GFP⁺ cells at the RP stage (*Figure 5G–G'1*). Co-transfection of Smad6 and RARα403

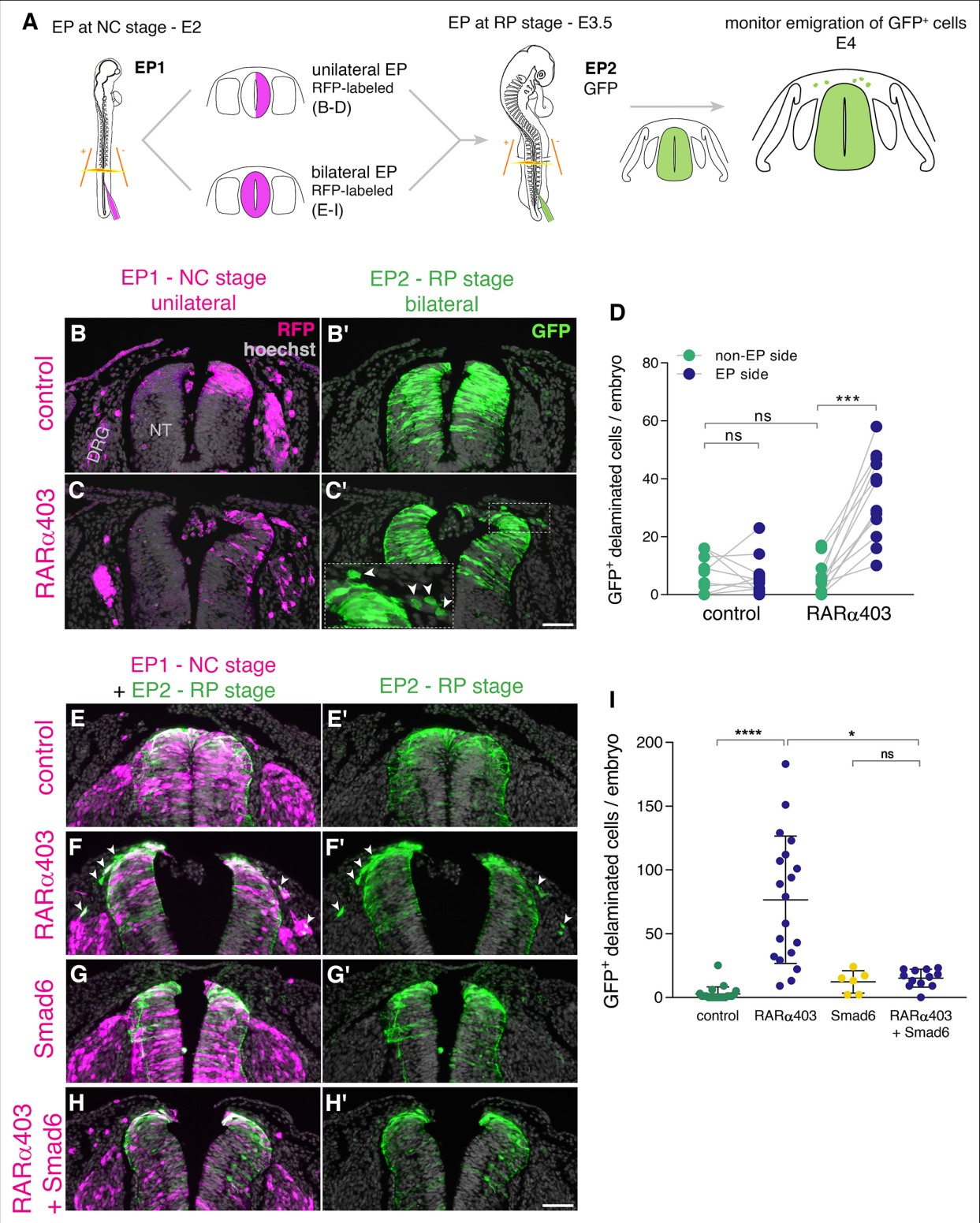

**Figure 5.** RA is responsible for the end of NC delamination. (**A**) Schematic representation of the experimental design for detection of late-emigrating cells (see text for details). (**B–D**) Embryos electroporated unilaterally at E2 (22-23ss) with either control or RARα403 plasmids, followed by a second bilateral electroporation at E3.5 with control GFP. Emigration of GFP+ cells was compared between electroporated and non-electroporated sides of control and experimental embryos at E4 (**D**). RFP was delivered with the early electroporation to monitor transfection efficiency. Images presented are adjacent to the rostral somitic domain. Whereas few GFP+ cells delaminated at the RP stage in controls, C' and inset show late-delaminating GFP+ cells ipsilateral to the early RARα403 electroporation (arrowheads). (**D**) Quantification of GFP+ cells that emigrated from either intact or electroporated (EP)

*Figure 5 continued on next page*

*Figure 5 continued*

sides of control and experimental embryos. Seventy two sections per embryo were analyzed at somitic levels 23–25. N = 10 and 11 embryos for control and RARα403 groups, respectively. Statistical tests applied were exact p-value Wilcoxon signed ranks paired test to compare between EP and non-EP sides of embryos inside each experimental group, and exact P-value Mann-Whitney test to compare between non-EP sides of control and RARα403 groups. ***p < 0.001. (**E–I**) Embryos were electroporated bilaterally at E2 (22-23ss) with control, RARα403, Smad6, or RARα403+ Smad6 constructs along with RFP. At E3.5, bilateral electroporations with GFP were performed and embryos were analyzed for emigration of GFP⁺ cells at E4. Images presented are opposite a rostral somitic domain. Note the presence of emigrated RFP⁺ cells in all treatments. Notice as well the presence of GFP⁺ delaminated cells in F,F', while virtually no GFP⁺ cells are seen outside of the NT in E,E', G,G' and H,H'. (**I**) Quantification of delaminated GFP⁺ cells. Seventy sections per embryo were analyzed at somite levels 23–25. N = 21, 19, 6 and 12 embryos were monitored for control, RARα403, Smad6, and Smad6+ RARα403 groups, respectively. *p < 0.05, ****p < 0.0001, approximate p-value Kruskal-Wallis with post-hoc Dunn's test. Abbreviations, ns, not significant; NT, neural tube, DRG, dorsal root ganglion. Scale bar, 50 µm.

The online version of this article includes the following source data and figure supplement(s) for figure 5:

**Source data 1.** RA is responsible for the end of NC delamination.

**Figure supplement 1.** Late-delaminating cells display NC characteristics.

**Figure supplement 1—source data 1.** Late-delaminating cells display NC characteristics.

**Figure supplement 2.** Gain of RA function fails to prematurely downregulate BMP signaling and NC traits.

**Figure supplement 2—source data 1.** Gain of RA function fails to prematurely downregulate BMP signaling and NC traits.

**Figure supplement 3.** No significant effect of VP16-RARαmisexpression on completion of NC emigration.

**Figure supplement 3—source data 1.** No significant effect of VP16-RARα misexpression on completion of NC emigration.

inhibited the RARα403-dependent extension of cellular EMT (*Figure 5H–H'1*). Together, these results suggest that late RA signaling ends the period of cell emigration in a BMP-dependent manner.

Next, we examined whether the late-delaminating cells exhibit a NC identity. For this purpose, embryos double-electroporated as described in *Figure 5A* were immunostained for the NC marker HNK1. While in control embryos no GFP-labeled cells were apparent outside the NT, and HNK1 stained only NC derivatives such as dorsal ganglia and ventral roots, in RARα403-treated embryos a considerable number of GFP⁺HNK1⁺ cells was also detected in the mesenchyme near the NT (*Figure 5—figure supplement 1A–B''*, arrowheads in B-B"). Remarkably, 95% ± 0.8% of the late-delaminating GFP-labeled cells were HNK1⁺ (*Figure 5—figure supplement 1C*), confirming their NC nature.

In addition, we examined the pattern of *foxd3*, which is expressed both in premigratory and migrating NC cells. Embryos were electroporated with RARα403 along with GFP at E2.5 and analyzed at E4. Control NTs were devoid of *foxd3* mRNA in the RP and mRNA signal was only detected in NC derivatives and dl2/V1 interneurons. In RARα403-treated embryos *foxd3* expression was evident in the dorsal NT (see also *Figure 3*), and also in NC cells that presumably emigrated recently (*Figure 5—figure supplement 1D-E'*, arrowheads), further confirming their NC identity.

Thus, local RA inhibition in the NT extends the period of NC production and further delamination into the RP stage. Collectively, these results point to RA being a major regulator of the end of the NC stage.

## Gain of RA signaling in the NT does not cause premature downregulation of BMP signaling or end of NC EMT

As precedently shown, the dorsal NT at the NC stage is characterized by the presence of BMP activity, expression of various NC-specific genes and NC emigration (*Figures 2, 3 and 5*). Likewise, RA activity is also apparent in NT at this stage (*Figure 1N*), suggesting that NC production and emigration are compatible with the endogenous level of RA signaling observed. Since repressing intrinsic RA activity prolongs NC-specific behaviors, we examined whether gain of RA function beyond the normal level of activity could inhibit prematurely the occurrence of NC traits. To this end, we electroporated a constitutively active form of human RAR-alpha fused to the transcriptional activator domain of VP16 (VP16-RARα) that activates RA target genes in a ligand-independent manner (*Novitch et al., 2003*). VP16-RARα significantly enhanced RA signaling, as measured with a short-lived RARE reporter (*Figure 5—figure supplement 2A-C*). Next, we monitored BMP activity (pSmad1/5/9), expression of Sox9 protein and *foxd3* mRNA in embryos fixed at advanced stages of NC production prior to the normal downregulation of the above. The number of sections or segments with pSmad, Sox9, or *foxd3* expression did not differ between control and VP16-RARα-treated cases (*Figure 5—figure*

*supplement 2D-N*, see details in legend). Next, the extent of NC emigration was reduced by 19% (p = 0.065, not significant) in VP16-RARα compared to control GFP-transfected NTs. Although this could reflect a tendency to a premature end of NC EMT, this limited effect could also be accounted for by death of some VP16-RARα–transfected cells, frequently observed in the lumen of the experimental cases, but not in controls (*Figure 5—figure supplement 3*). Together, whereas RA signaling is necessary for the end of NC EMT and associated events, it is not sufficient to prematurely end these processes. This could suggest that it is not the sole activity of RA in the NT that causes the end of NC EMT but rather an interplay between RA signaling in the context of a gene network that changes between NC and nascent RP stages.

## Inhibition of RA signaling downregulates only a subset of RP markers

A major question that remains to be addressed is whether the end of the NC stage is sufficient for signaling the formation of the definitive RP, or whether these are two independently regulated processes. Given that RA deficiency delays the end of the NC stage, we asked whether this occurs at the expense of timely RP development.

As recently shown (*Ofek et al., 2021*), there are differences in the gene expression profile between premigratory NC cells and definitive RP cells. We selected a number of RP-specific genes that are not expressed in the dorsal NT during the NC stage and examined the effect of RA downregulation on their expression. As described in *Figure 2*, expression of RP-specific BMP inhibitors was strongly reduced. Likewise, mRNA encoding *Slit1*, a chemorepellent of post-floor plate commissural axons (*Kidd et al., 1999*) was also decreased under experimental conditions (*Figure 6—figure supplement 1*). In contrast, additional RP markers, such as *Rspo1*, *NDP (norrin)*, or *draxin*, were normally expressed (*Figure 6—figure supplement 1*), suggesting that development of a number of RP traits is independent of RA signaling.

Since both NC and RP markers are present at the RP stage upon inhibition of RA signaling (*Figure 3*, *Figure 6—figure supplement 1*), we proceeded to clarify whether NC and RP markers are expressed in the same cells, or, alternatively, whether there is a mix of different cell types in the RP of RARα403-treated embryos. First, we performed ISH of *Rspo1* (RP marker) and *foxd3* (NC marker) on adjacent sections of RP-stage embryos electroporated at E2.5 with either RARα403 or control GFP (*Figure 6*). Whereas in control NTs, only *Rspo1* was apparent in the RP, both *Rspo1* and *foxd3* were detected in an overlapping pattern in the RP of the treated embryos (*Figure 6A–B'''*).

In addition, we combined a fluorescent ISH of *Rspo1* with immunostaining for SNAI2 or Sox9 in the same sections. Control NTs exhibited SNAI2 and Sox9[+] nuclei only outside the *Rspo1*[+] RP (*Figure 6C–C'' and E–E''*, see also *Figure 3*). In striking contrast, both markers were detected in the same dorsal NT cells in the absence of local RA signaling (*Figure 6D–D'', F–G'', H1*). In some instances, *Rspo1*[+]Sox9[+] cells were also noticed undergoing delamination from the NT (*Figure 6G–G''*, arrowhead).

Next, we examined whether gain of RA function implemented at the NC stage, is able to cause a premature onset of expression of selected RP genes. Serial section analysis, performed in regions flanking the beginning of gene expression, revealed that the onset of both *BAMBI* and *Rspo1* mRNAs in the NT was similar in control and VP16-RARα-treated embryos (*Figure 6—figure supplement 2*).

Together, these data show that the loss of RA signaling does not totally prevent the advent of RP markers and, reciprocally, high levels of RA signaling at the NC stage are not sufficient to promote a premature advent of RP-specific genes. Hence, loss of NC identity and gain of RP fate are independently-regulated processes. Nevertheless, timely RA activity in the dorsal NT is required for the proper segregation of NC from RP fates.

## RP-derived dI1 interneuron development only partly depends on RA signaling

We further investigated whether depletion of RA signaling had any effect on the role of the RP as a signaling center for dorsal interneuron development. We immunostained RARα403 and control-electroporated embryos with the dI1 interneuron marker BarHL1, and analyzed the number of BarHL1[+] cells and their spatial distribution. Whereas no significant difference in the number of BarHL1[+] interneurons could be monitored between control and treated embryos (*Figure 7A*), there was a collective dorsal shift in their position in the RARα403-treated NTs, with some of BarHL1[+] cells residing in the

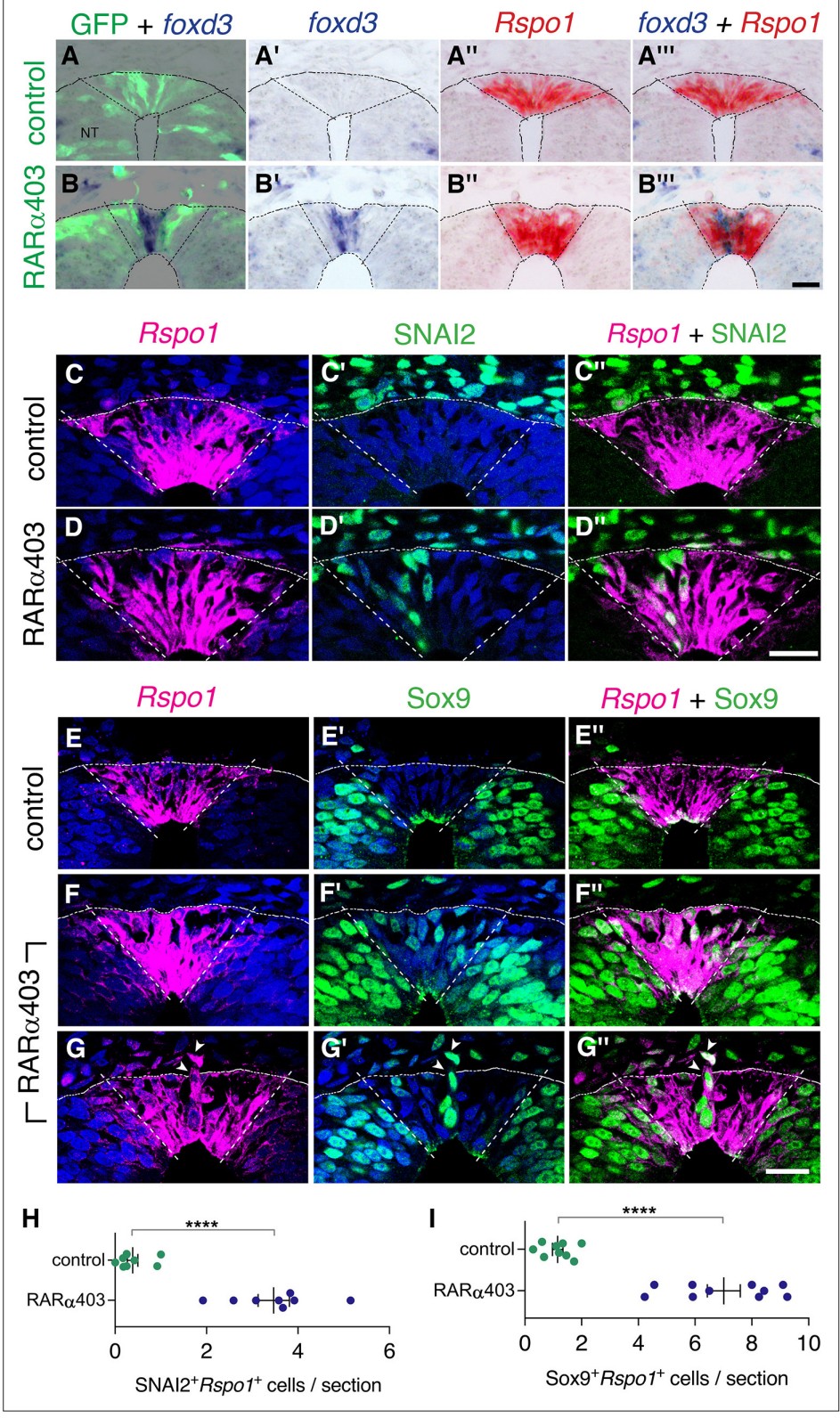

**Figure 6.** Overlapping expression in RP of both NC and RP markers upon inhibition of RA signalling. (**A-B'''**) Embryos co-electroporated at E2.5 (27ss) with either control PCAGG or RARα403 along with a GFP plasmid. Embryos were fixed at E4, and adjacent sections were in-situ hybridized for *foxd3* or *Rspo1* and then superimposed to reveal overlapping domains of gene expression. Dotted lines delineate the *Rspo1* expression domain. In control

*Figure 6 continued on next page*

*Figure 6 continued*

embryos, no *foxd3* expression was detected in the RP (N = 2, and see also **Figure 3**). In RARα403-treated embryos, overlapping expression was detected in all four embryos examined. Imaging and analysis were performed somite levels 24–26. (**C-D"**) Immunostaining for SNAI2 and fluorescent ISH for *Rspo1* were combined on the same sections of embryos electroporated at E2.5 (27ss) with either control PCAGG or RARα403 and analyzed at E4. Dashed lines mark the *Rspo1* expression domain. Note the presence of SNAI2⁺*Rspo1*⁺ cells in the RP in D". (**E-G"**) Immunostaining for Sox9 and fluorescent ISH for *Rspo1* were performed as above. Note the presence of Sox9⁺*Rspo1*⁺ cells in the RP in F" and G". Additionally, delaminating Sox9⁺*Rspo1*⁺ cells (arrowheads in G-G") were apparent under experimental conditions. (**H,I**) Quantification of SNAI2⁺*Rspo1*⁺ and Sox9⁺*Rspo1*⁺ cells in the RP. Imaging and analysis were performed at somite levels 24–26. N = 8 and 8 for SNAI2 in controls and RARα403. N = 9 and 10 embryos for control and RARα403 groups stained with Sox9, respectively. ****p < 0.0001, Welch's t-test. Abbreviations, NT, neural tube. Scale bar, 20 μm.

The online version of this article includes the following source data and figure supplement(s) for figure 6:

**Source data 1.** Overlapping expression in RP of both NC and RP markers upon inhibition of RA signalling.

**Figure supplement 1.** Inhibition of RA signaling only partially affects expression of definitive RP markers.

**Figure supplement 1—source data 1.** Inhibition of RA signaling only partially affects expression of definitive RP markers.

**Figure supplement 2.** Gain of RA signaling does not cause a premature upregulation of RP-specific genes.

**Figure supplement 2—source data 1.** *Figure 6—figure supplement 2* – Gain of RA signaling does not cause a premature upregulation of RP-specific genes.

---

RP domain itself (**Figure 7B–E'**, arrowheads). This shift did not impact the overall size of the RP, as the area of the *Rspo1*-positive RP did not significantly differ between control and treated embryos (control, 1672 ± 121.3 μm² (n = 8); compared to RARa403, 1790 ± 146.1 μm² (n = 8), p = 0.544, student's t-test).

Thus, whereas inhibition of RA signaling does not affect RP-mediated formation of dI1 interneurons, it interferes with the spatial information conveyed onto dorsal interneurons from the RP, likely by impairing the formation of the boundary between the RP and more ventral domains. These findings further indicate that RA signaling not necessarily influences development of RP properties but rather impairs temporal and spatial segregation between dorsal cell lineages.

Data from young caudal neural plate explants and from VAD embryos that lack RA, showed that early RA signaling from developing somites is required for dorso-ventral patterning of the NT, for the specification of ventral cell types (motoneurons and V1, V2 interneurons) and for neuronal differentiation (**Diez del Corral et al., 2003**; **Sockanathan and Jessell, 1998**; **Wilson et al., 2004**). We therefore examined whether later inhibition of RA activity as performed in the present context merely concerns the dorsal NT or, alternatively, has a general effect on dorso-ventral patterning. To this end, we examined the effects of bilateral RA attenuation on the expression of Pax7 in the dorsal NT, and of the motoneuron-specific protein Hb9. RARα403 caused a mild 12% increase in the area of expression of Pax7 when compared to controls (**Figure 7—figure supplement 1A-C**), and had no effect on either the localization or extent of expression of Hb9 (**Figure 7—figure supplement 1D-F**). These data corroborate that the effects documented are directed to the dorsal NT and do not result from overall changes in its dorso-ventral patterning.

## The dorsal NT as the source of RA responsible for NC to RP transition

As shown in **Figure 1** and **Figure 1—figure supplement 1**, the dorsal NT becomes a source of RA during the period marking the end of NC emigration and transition to the RP stage, suggesting this is the local source of RA that underlies the effects demonstrated here. Yet, at the same time, RA is still produced in the adjacent paraxial mesoderm. To begin addressing whether the latter tissue accounts for the effects observed, we separated the paraxial mesoderm from the NT with an impermeable membrane and assessed the development of the RP. A unilateral slit was performed at the epithelial somite stage into which a piece of aluminum foil was inserted (**Figure 8A**). We analyzed RP formation using *BAMBI* expression as a sensitive readout for normal RP ontogeny that is downregulated in the absence of RA signaling (**Figure 2**). Notably, *BAMBI* remained strongly expressed in the RP of treated embryos to the same extent as the control sides, despite separation from the mesoderm and virtual disappearance of the somitic tissue that lost contact with the NT (**Figure 8A–C'**). To

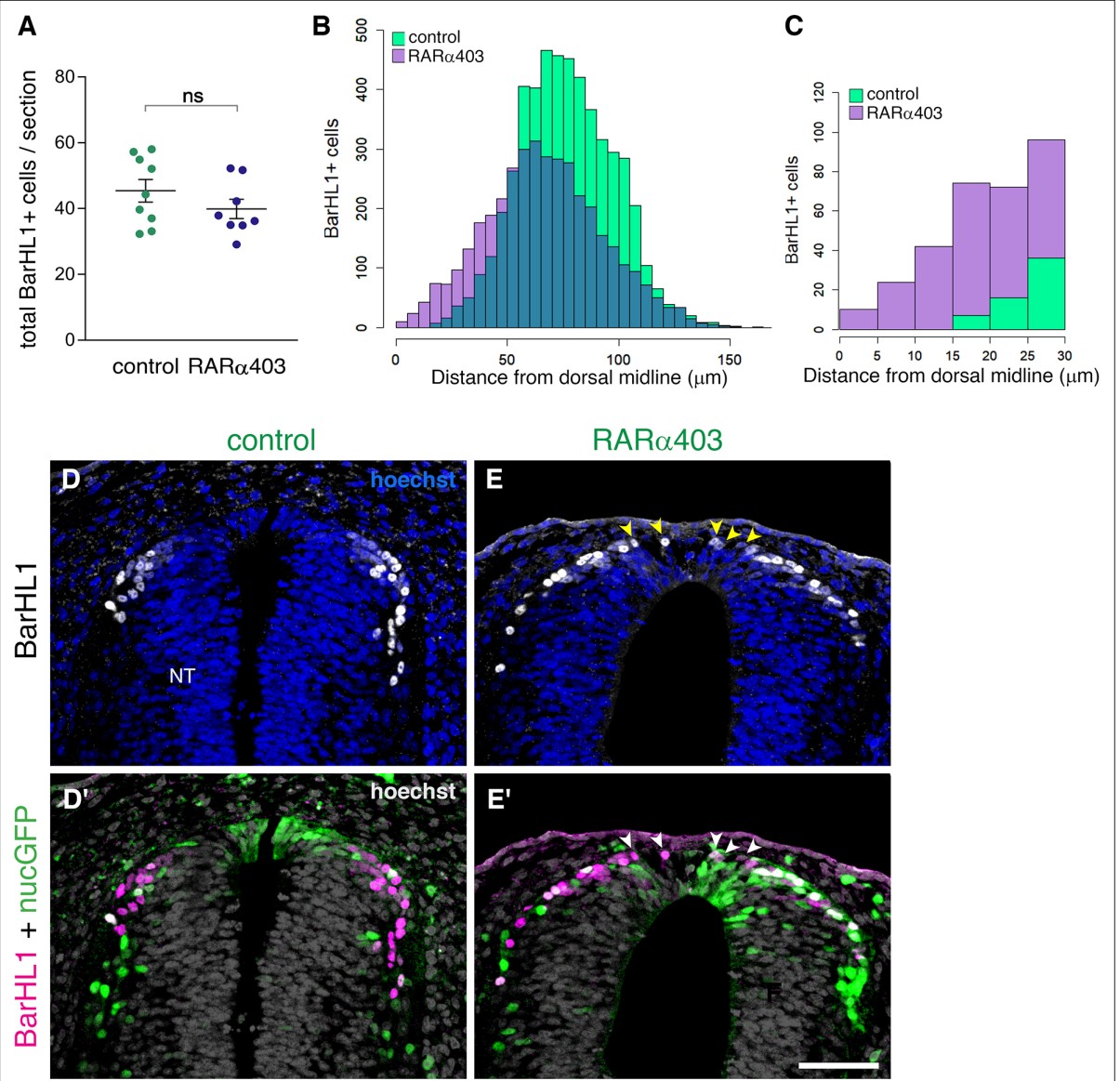

**Figure 7.** RP-derived dI1 interneuron development only partly depends on RA signalling. (**A**) Quantification of BarHL1+ dI1 interneurons in E4 embryos, electroporated at E2.5 (27ss) with either control PCAGG or RARα403. No significant difference in the number of BarHL1+ neurons was monitored. Ten-to-14 sections were analyzed per embryo; N = 9 and 8 embryos for control and RARα403 groups, respectively. Student's unpaired t-test was applied. (**B, C**) Spatial distribution of BarHL1+ cells measured as the distance from the dorsal midline. 5126 cells were counted in the control group (green), and 3886 in the RARα403 group (purple). The area of overlap of control and experimental cells is in blue. Note the dorsal shift of cells in the RARα403-treated group when compared to controls. In (**B**) all the cells are included, while in (**C**) only cells included in a 30 μm distance from the dorsal midline, comprising the RP region, are presented. (**D-E'**) Confocal images of BarHL1-stained sections. Nuclear GFP (nucGFP) marks the electroporated domain. Note in E and E' the presence of BarHL1+ interneurons inside the RP domain (arrowheads) compared to controls in which these neurons are localized ventral to the RP. Nuclei are visualized with Hoechst. Abbreviation, NT, neural tube. Scale bar, 50 μm.

The online version of this article includes the following source data and figure supplement(s) for figure 7:

**Source data 1.** RP-derived dI1 interneuron development only partly depends on RA signaling.

**Figure supplement 1.** Loss of RA signaling during NC to RP transition does not affect dorso-ventral patterning of the NT.

**Figure supplement 1—source data 1.** Loss of RA signaling during NC to RP transition does not affect dorso-ventral patterning of the NT.

exclude the possibility that RA from the mesoderm of the non-separated side is sufficient to induce normal RP development, a slit was made along the dorsal midline of the NT following insertion of the barrier, thus opening the NT and creating a physical gap between both halves of the neuroepithelium (*Figure 8D*). A comparable expression of *BAMBI* mRNA on both hemi-RPs was still apparent under

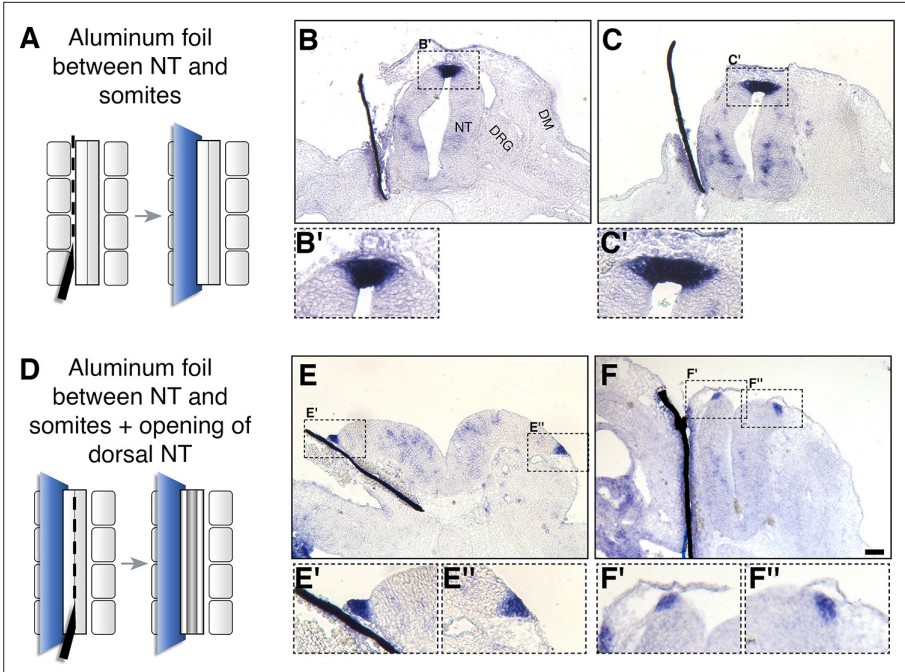

**Figure 8.** Somitic RA is dispensable for NC-to-RP transition. (**A, D**) Schematic representations of the experimental design to mechanically separate the NT from somitic mesoderm using an impermeable aluminum foil inserted at 27ss at the level of the rostral segmental plate and five newly formed somites. Embryos were fixed at E4. (**B-C',** **E-F'**) ISH for *BAMBI* on sections of embryos with intact (**B-C'**) or split (**E-F''**) dorsal NTs following grafting of a unilateral barrier between NT and somites. Representative images from two different embryos are presented for each treatment. *BAMBI* expression remained unchanged in the RP opposite to the operated sides in 7/7 embryos with an intact RP, and in 4/4 embryos with a split RP. Note as well that distal to the barriers, most of the somitic mesoderm disappeared (**B,C,E**). In some instances, the somatic layer of the lateral plate mesoderm approached the barrier (**F**). Abbreviations, DM, dermomyotome, DRG, dorsal root ganglion, NT, neural tube. Scale bar, 50 μm.

these conditions (*Figure 8E–F''*), suggesting that the paraxial mesoderm is not the source of RA important for NC to RP transition.

Next, we asked what restricts expression of *Raldh2* to the RP. *Foxd3, snai2,* and *sox9* are transcribed only during the NC stage in a complementary pattern vis-a-vis *Raldh2* (*Krispin et al., 2010b*; *Ofek et al., 2021*). In addition, RP-derived RA inhibits expression of these three transcription factors (*Figure 3*). We therefore predicted that the onset of *Raldh2* production in RP could be initiated by the normal downregulation of these NC-specific genes. To test this notion, we extended the activity period of *foxd3, snai2,* or *sox9* by misexpressing each of them close to their time of disappearance from the dorsal NT (30ss at the flank level)(*Krispin et al., 2010b*) and monitored *Raldh2* mRNA at the RP stage (E4). Each of the factors cell-autonomously prevented the onset of *Raldh2* expression in the RP when compared to control GFP-electroporated cases (*Figure 9*). Hence, expression of NC (*foxd3, snai2, sox9*) and RA-associated RP traits (e.g; *Raldh2*) stand in a mutually repressive temporal relationship. Together, these interactions provide novel insights into the existence of a gene regulatory network responsible for the transition between NC and RP (*Figure 10*).

## Discussion

Here, we show that dorsal NT-derived RA controls the end of NC production and emigration (*Figure 10*). This effect is mediated through BMP signaling, at least via the upregulation of BMP inhibitors in the nascent RP, which prevents persistent BMP activity and ensures the timely cessation of NC production (*Figure 10*; *Nitzan et al., 2016*). We previously reported that BMP produced in the dorsal NT is an essential inducer of NC EMT and emigration; a graded activity of BMP is created along the rostro-caudal axis of the embryo thanks to a counter-gradient of its inhibitor noggin. By the time of initial somite dissociation, noggin expression is downregulated by somite derived factors (*Sela-Donenfeld*

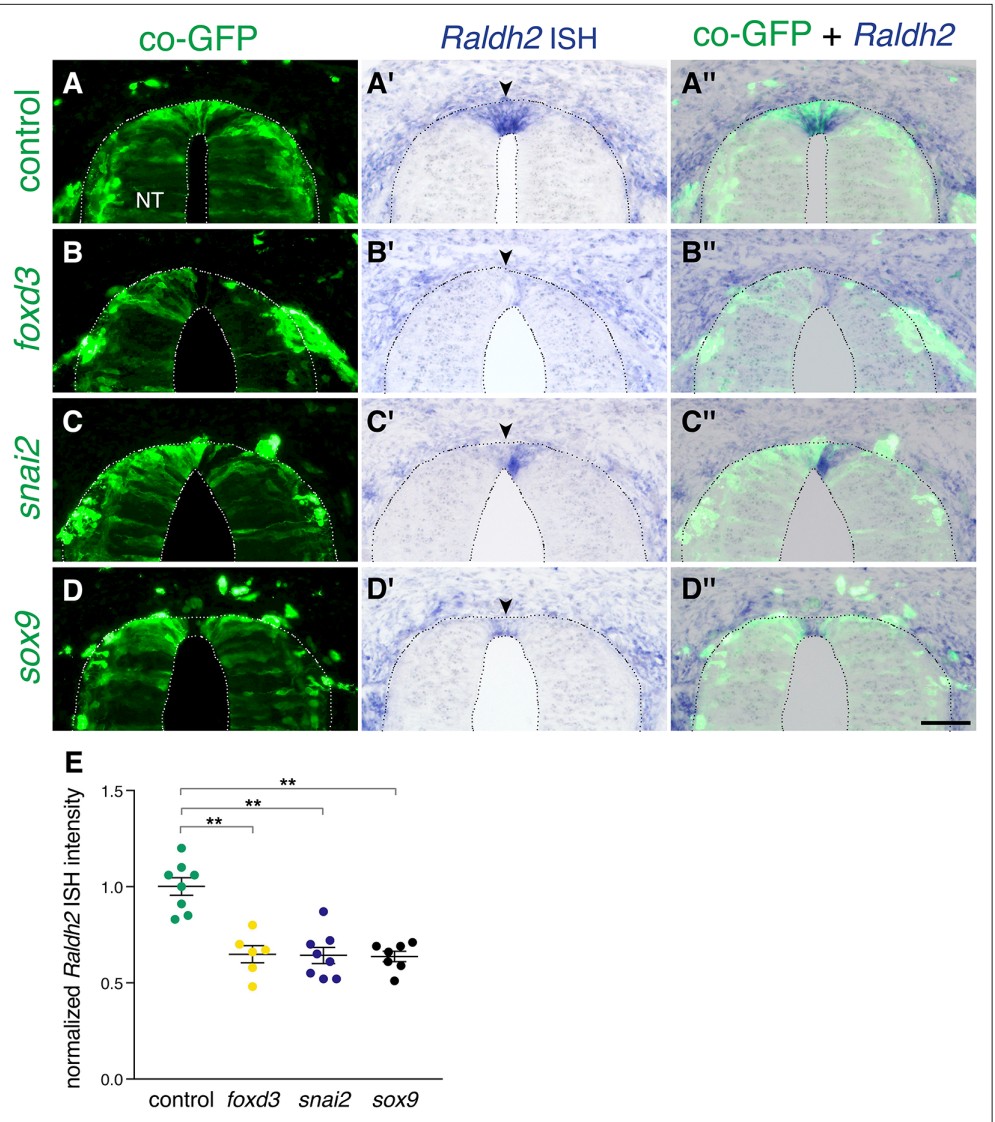

**Figure 9.** Extension of the activity of NC-specific genes prevents the onset of expression of *Raldh2* in the nascent RP. (**A-D''**) Embryos were electroporated at E2.5 (27ss) with control unlabeled PCAGG vector or with *foxd3*, *snai2* or *sox9*-expressing constructs along with GFP, and analyzed at E4 for expression of *Raldh2*. Note the lack of *Raldh2* expression in the transfected cells of all treated groups compared to controls (arrowheads). (**E**) Quantification of *Raldh2* ISH intensity. Six to 30 sections were analyzed at somite levels 24–26. N = 8,6,8 and 7 embryos for control, *foxd3*, *snai2*, and *sox9* groups, respectively. **p < 0.01 via Kruskal-Wallis with post-hoc Dunn's test. Abbreviations, NT, neural tube. Scale bar, 50 μm.

The online version of this article includes the following source data for figure 9:

**Source data 1.** Extension of the activity of NC-specific genes prevents the onset of expression of *Raldh2* in the nascent RP.

---

*and Kalcheim, 1999*; *Sela-Donenfeld and Kalcheim, 2002*), later found to be accounted for by an interplay between mesodermal RA and Fgf8 (*Martínez-Morales et al., 2011*). This group reported that during gastrulation, RA is required for NC specification, as revealed by analysis of VAD quail embryos. Next, during somite formation, somite-derived RA is necessary for the onset of emigration of specified NC progenitors but at advanced somite stages it is dispensable for the subsequent maintenance of cellular EMT (*Martínez-Morales et al., 2011*). Together, these and the present results highlight a rather complex behavior of RA at four sequential stages of NC ontogeny, specification, onset, maintenance and completion of EMT.

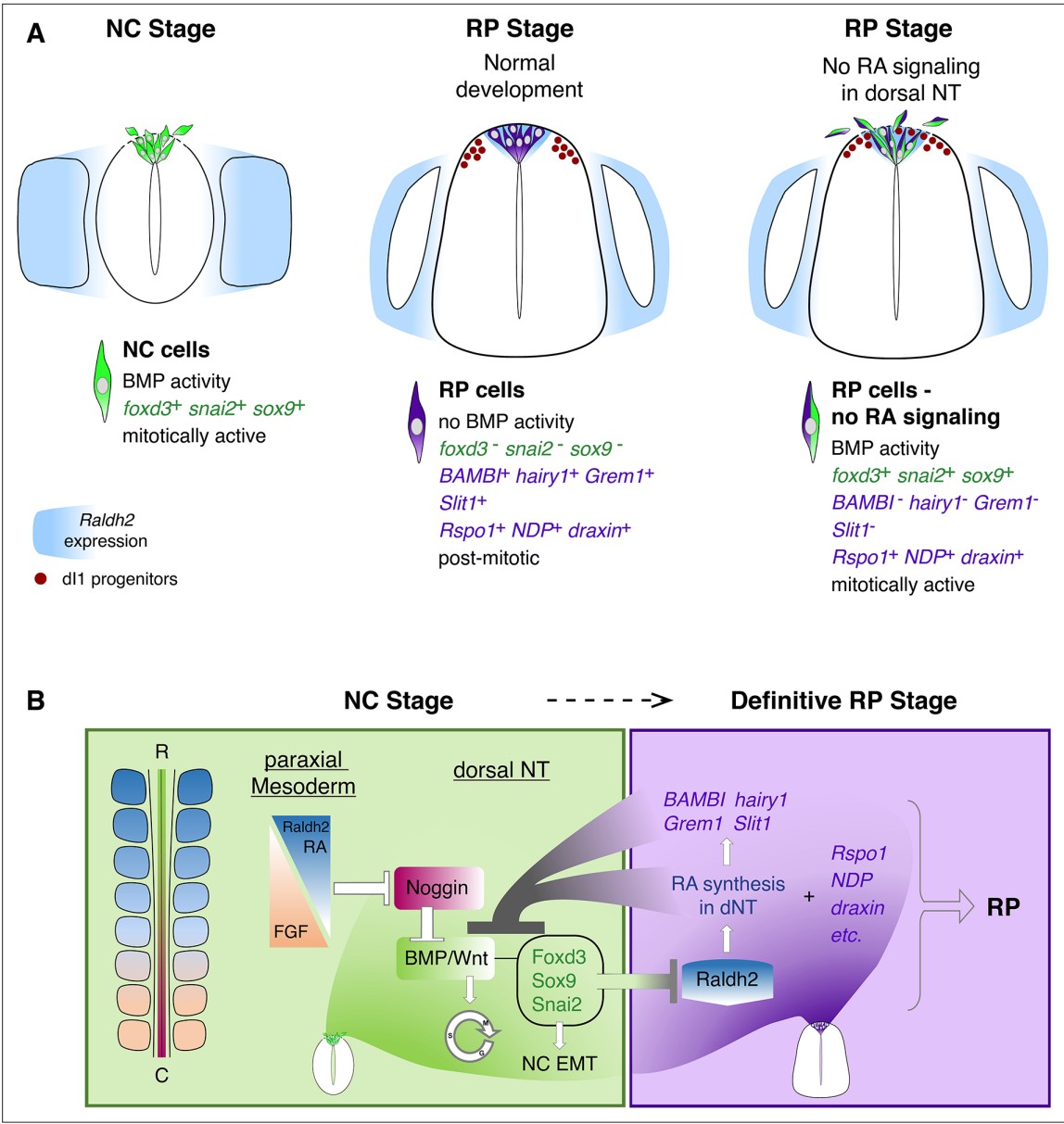

**Figure 10.** Schematic diagram summarizing the role of RA in the transition from NC to RP. (**A**, left panel) At the NC stage, premigratory progenitors residing in the dorsal NT (green) exhibit BMP signaling activity, express specific NC markers (e.g. *foxd3*, *snai2* and *sox9*, green) and proliferate extensively. Following a process of EMT, NC cells delaminate from the NT through a discontinuous basement membrane (dashed line) to migrate away. At the NC stage, RA is supplied to the NT from the paraxial mesoderm, where it is produced by Raldh2 (blue). (A, middle panel) Under normal conditions, upon cessation of NC production and emigration, a definitive RP forms. RP cells (purple) are refractory to BMP signaling, and exhibit a different gene expression profile when compared to NC cells: they downregulate *foxd3*, *snai2*, and *sox9* and upregulate RP-specific genes (purple), among which are known inhibitors of BMP signaling. RP cells gradually exit the cell cycle and do not delaminate, regenerating a continuous basement membrane and epithelial traits. During the transition between NC and RP stages, *Raldh2* expression begins in the dorsal NT (blue in RP), in addition to RA still produced in somites. The RP functions as a signaling center, regulating specification and/or differentiation of dorsal interneurons (red, dI1 progenitors) that localize ventral to the RP. (A, right panel) In the present study we demonstrate that inhibition of RA signaling in the dorsal NT, impairs the end of NC production and consequently, the normal transition between NC and RP stages. First, under experimental conditions, dorsal NT cells at the RP stage retain active BMP signaling likely via downregulation of BMP antagonists (*BAMBI, hairy1, Grem1*) otherwise selectively expressed in RP. Second, they express both NC and RP markers (illustrated as green/purple cells), Third, they functionally behave as NC cells, maintaining mitotic activity, impaired epithelial traits (such as a discontinuous basement membrane, dashed line), and continuous emigration from the NT. Fourth, although the advent of several RP-specific markers is normal (e.g; *Rspo1, NDP, draxin*) and the number of dI1 progenitors (red circles) is unchanged, the spatial segregation between NC, RP, and dI1 interneuron lineages is compromised. (**B**) A proposed model for the transition between NC (green) and RP (purple) stages. At early stages, reciprocal gradients of RA and FGF in the paraxial mesoderm result in downregulation of the BMP inhibitor Noggin

*Figure 10 continued on next page*

*Figure 10 continued*

in the dorsal NT. This allows for the activation of BMP and Wnt signaling pathways, which promote cell proliferation and induce NC EMT. As NC cells delaminate and leave the NT, NC-specific genes (*foxd3, sox9,* and *snai2*) are downregulated. Since the latter genes inhibit the synthesis of *Raldh2* in the nascent RP, their disappearance enables the onset of *Raldh2* expression allowing for the local synthesis of RA in RP. RA in turn inhibits BMP and consequently, Wnt signaling, either directly or via upregulation of BMP inhibitors (*BAMBI, hairy1, Grem1*). We propose the existence of mutual cross-inhibitory interactions between RA-responsive genes that characterize NC and RP stages, respectively, in the temporal sequence leading to the formation of the definitive RP and its segregation from NC.

Our findings are consistent with previous results reporting that RA repressed BMP signal in P19 embryonic carcinoma cells; in vivo, RA antagonized BMP-regulated differentiation and proliferation of neural progenitor cells in the NT (*Sheng et al., 2010*). Likewise, RA was also found to inhibit Wnt signaling (*Shaker et al., 2020*), that in the present context acts downstream of BMP and whose activity was extended in RP by loss of RA signaling.

In normal development, the end of NC emigration is followed by the advent of the definitive RP of the CNS (*Rekler and Kalcheim, 2021*). This is associated with a progressive withdrawal from the cell cycle, the regeneration of a complete basal lamina overlying the dorsal NT, of apico-basal properties of the RP epithelium and the corresponding downregulation and upregulation, respectively, of NC and RP-specific genes (*Nitzan et al., 2016*; *Ofek et al., 2021*). We report that, in the absence of RA activity, there is a continuity of several molecular and cellular properties of the NC well into the RP stage, including the presence of a subset of RP cells that co-expresses NC markers. Together, lack of local RA activity blurs the spatial and temporal segregation between NC and RP (*Figure 10A*).

Despite these phenotypes, it is notable that RP-specific markers such as *Rspo1*, *NDP*, or *draxin*, were not affected, suggesting that a RP is able to develop even if RA activity is inhibited. This raises the fundamental question whether the end of NC formation and the ensuing development of a RP are regulated by the same or by independent mechanisms. Our results would suggest that the mechanisms accounting for the cessation of NC phase are necessary but not sufficient to explain all aspects of RP development. Consistent with this notion, spatio-temporal lineage tracings revealed that RP segregate from neural progenitors of the NC before the latter complete emigration (*Krispin et al., 2010b*; *Nitzan et al., 2013*). In addition, gain and loss of Notch function in avians and mice, respectively, resulted in changes in RP formation with no apparent effects on NC cell behavior (*Ofek et al., 2021*). This would indicate that although NC and RP lineages arise from common progenitors (*Bronner-Fraser and Fraser, 1988*; *Nitzan et al., 2013*) by different exposure times and/or levels of BMP signaling (*Nitzan et al., 2016*; *Timmer et al., 2002*; *Tozer et al., 2013*), they segregate rather early from each other.

Among the RP genes whose expression is perturbed in the absence of RA signaling, we found *BAMBI*, *HES/hairy1* and *Grem1*, none of which is expressed in the dorsal NT at NC stages. These genes, whose transcription was prevented by treatment with RARα403 and/or Cyp26, encode inhibitors of BMP signaling. Furthermore, HES represses BMP activity and subsequent NC emigration (*Nitzan et al., 2016*). Hence, RA-sensitive aspects of RP development are linked to its regulation of BMP activity. Moreover, since RA-dependent downregulation of BMP activity accounts for only selected facets of RP ontogeny, we posit that RP development comprises both RA-dependent and RA-independent mechanisms.

We observed that maintenance of NC traits in RARα403-treated NTs did not interfere with differentiation of a normal number of RP-dependent dorsal interneurons of type 1, suggesting that the role of the RP as an interneuron inducer center was not altered. However, their topographical dorsoventral distribution was abnormal as instead of organizing into discrete nuclei, interneurons redistributed along the basal surface of the NT comprising the RP itself, further demonstrating that the structural integrity of this signaling center and/or of its boundaries is compromised. This is consistent with the observed local changes in epithelial architecture, cell cycle properties and basal lamina deposition and might be partially accounted for by the continued emigration of NC cells driving a dorsal shift of interneurons. Along this line, cell shape changes associated with continuous mitosis were reported to impact tissue architecture, including epithelial polarity and cellular EMT (*Despin-Guitard and Migeotte, 2021*).

In addition, since dl1 interneurons are also specified by BMP signaling (*Chesnutt et al., 2004*; *Hegarty et al., 2013*; *Timmer et al., 2002*), it is conceivable to speculate that maintenance of BMP

responsiveness in the RP of RARα403-treated NTs affects the normal dorso-ventral gradient of BMP activity (*Tozer et al., 2013*), perhaps eliciting indirectly an attractive function that enables dI1 interneurons to migrate into the RP domain. In this context, RP-derived BMP exhibits at a later stage chemorepellent properties vis-à-vis commissural axons (*Augsburger et al., 1999*; *Butler and Dodd, 2003*), putting forward the additional possibility of guidance functions for this morphogen under the present experimental conditions. One of the genes downregulated upon abrogation of RA function is *Slit1*, a known repellent of axonal guidance (*Kidd et al., 1999*), as well as of cell migration (*Halperin-Barlev and Kalcheim, 2011*; *Theveneau and Mayor, 2012b*). This is consistent with recent data pointing to an indirect role for RA in axonal guidance (*Isabella et al., 2020*). Loss of *Slit1* in the absence of RA activity could also account for the dorsal shift of dI1 interneurons into the RP domain, and possible interactions between the three factors in this context remain to be explored.

As discussed precedently, the NT is sensitive to RA throughout development, with RA playing various functions at different stages [this study, *Diez del Corral and Morales, 2014*; *Martínez-Morales et al., 2011*; *Wilson et al., 2004*]. Hence, it is plausible that the mechanisms mediating these effects differ as a function of time and context. Here we report that precisely before the end of NC delamination, RA begins to be synthesized in the dorsal NT (e.g; *Raldh2* expression) in addition to the paraxial mesoderm, raising the notion that this new, local source of RA accounts for the end of NC development. Notably, *Raldh2* is also synthesized in the RP of mouse embryos (*Ofek et al., 2021*). Unfortunately, available mutants of *Raldh2* are embryonic lethal at about E8.75 (*Niederreither et al., 1999*), precluding a focal analysis of the transition from NC to RP. In this sense, the avian embryo provides an excellent model to address events restricted both in time and space. The first is the activity of somite vs. RP-derived RA. Two methods we implemented to abrogate RA signaling in the dorsal NT that act primarily in a cell autonomous fashion. However, it is still possible that given the secreted nature of the ligand, Cyp26A1 and/or RARα403 misexpression in the dorsal NT also provide a sink for somitic RA. Despite robust separation of the mesoderm from the NT, expression of *BAMBI*, a most sensitive and significant readout of RA activity, was not altered, suggesting that mesodermal RA is dispensable. What accounts for this shift in responsiveness from a mesodermal towards a neural source of factor? First, local RA could provide a more efficient signal to target progenitors given the increasing size of the embryo and the development of a continuous basement membrane dorsal to the NT. Second, cofactors of RA such as *CRABP1*, could change the nature of the cellular response as it only appears at the RP stage. Together, changes in the molecular landscape between NC and RP stages (*Ofek et al., 2021*), that partly depend on local RA activity, may alter the nature of the interactions between RA and BMP/Wnt to signal a timely cessation of NC production.

Second, we asked what restricts the synthesis of *Raldh2* to the dorsal NT at the RP stage. Whereas RP-derived RA inhibits NC-specific traits including transcription of *foxd3*, *sox9*, and *snai2*, each of these factors in turn represses the onset of *Raldh2* transcription in the nascent RP. Thus, as long as NC genes are expressed in the early dorsal NT (NC stage), local *Raldh2* and consequent RA synthesis does not take place. Hence, a cross-repressive interaction exists between NC and RP-specific genes downstream of RA, an emerging temporal property of the network that determines the segregation of sequential cell lineages (*Figure 10B*).

Our study thoroughly documents a role of local RA activity on the end of NC production and ensuing RP architecture. A comprehensive elucidation of the molecular mechanism/s responsible for inhibition of BMP signaling by local RA is the next obligatory step. In this regard, different RA enhancers might be expressed at either stage and be regulated by separate factors. For example, a specific enhancer driving expression of *Raldh2* is activated only at the definitive RP stage (*Castillo et al., 2010*). This enhancer contains Tcf binding sites and thus may be activated by Wnt signaling. In turn, RP-derived Raldh2 and resulting RA could negatively feedback on Wnt signaling in the formed RP either directly or through BMP acting upstream of Wnt (*Figure 10B*). Consistent with this possibility, we presently report that RP-derived RA signaling induces expression of the BMP inhibitors *BAMBI*, *Grem1* and *hairy/hes* specifically in RP, and this can be one mechanism whereby RA represses BMP signaling. Along this line, we previously showed that misexpression of *Hes1* at the NC stage downregulates BMP activity and represses NC EMT (*Nitzan et al., 2016*). Furthermore, RA could repress BMP signaling by inactivating Smad proteins via ubiquitination, as shown to be the case in selected cell lines (*Sheng et al., 2010*). Altogether, the present study provides an initial mechanistic

explanation for the completion of NC production and separation between PNS (NC) and CNS (RP and dI1) lineages during development.

# Materials and methods

## Key resources table

| Reagent type (species) or resource | Designation | Source or reference | Identifiers | Additional information |
|---|---|---|---|---|
| Strain, strain background (Japanese quail) | *Coturnix coturnix japonica* | Moshav Mata | NCBI taxon: 93,934 | |
| Antibody | anti-GFP (Rabbit polyclonal) | Invitrogen, Thermo-Fisher Scientific | Cat#A6455; RRID:AB_221570 | IF(1:1000) |
| Antibody | anti-GFP (Mouse monoclonal) | Abcam | Cat#Ab38689; RRID:AB_732715 | IF(1:100) Not available anymore |
| Antibody | anti-RFP (Rabbit polyclonal) | Acris | Cat# AP09229PU-N; RRID:AB_2035909 | IF(1:1000) |
| Antibody | anti-pSmad1/5/8 (guinea pig polyclonal) | Ed Laufer | N/A | IF(1:300) |
| Antibody | anti-pSmad1/5/9 (Rabbit monoclonal) | Cell Signaling Technology | Cat#CST13820; RRID:AB_2493181 | IF(1:500) |
| Antibody | anti-H3-pS10 (Mouse monoclonal) | Abcam | Cat#Ab14955; RRID:AB_443110 | IF(1:400) |
| Antibody | anti-Laminin (Rabbit polyclonal) | Sigma, Israel | Cat#L9393; RRID:AB_477163 | IF(1:100) |
| Antibody | anti-Sox9 (Rabbit polyclonal) | Millipore | Cat#AB5535; RRID:AB_2239761 | IF(1:150) |
| Antibody | anti-SNAI2 (Rabbit monoclonal) | Cell Signaling Technology | Cat#CST9585; RRID:AB_2239535 | IF(1:500) |
| Antibody | anti-CD57(HNK1) (Mouse monoclonal) | BD Biosciences | Cat#559048; RRID:AB_397184 | IF(1:500) |
| Antibody | anti-BarHL1 (Rabbit polyclonal) | Sigma, Israel | Cat# HPA004809; RRID:AB_1078266 | IF(1:300) |
| Antibody | anti-Pax7 (Mouse monoclonal) | DHSB | Cat# pax7; RRID:AB_528428 | IF(1:10) |
| Antibody | anti-Hb9 (mouse monoclonal) | DHSB | Cat# 81.5C10; RRID:AB_2145209 | IF(1:200) |
| Recombinant DNA reagent | pCAGG (plasmid) | *Krispin et al., 2010b* | | |
| Recombinant DNA reagent | pCAGGS-EGFP (plasmid) | *Krispin et al., 2010b* | | |
| Recombinant DNA reagent | pCAG-mGFP (plasmid) | Addgene | RRID:Addgene_14757 | |
| Recombinant DNA reagent | pCAGGS-RFP (plasmid) | *Ofek et al., 2021* | | |
| Recombinant DNA reagent | pCAGGS-RARα403 (plasmid) | This paper | | Subcloned as described in Methods section |
| Recombinant DNA reagent | pCAGGS-Cyp26A1 (plasmid) | This paper | | Subcloned as described in Methods section |
| Recombinant DNA reagent | pCAB-cSmad6 (plasmid) | *Nitzan et al., 2016* | | |
| Recombinant DNA reagent | pCAG-VP16-RARα-IRES-eGFP (plasmid) | *Novitch et al., 2003* | | From S. Sockanathan |
| Recombinant DNA reagent | pGL3-RARE-SV40-AP (plasmid) | *Gupta and Sen, 2015* | | From J. Sen |

*Continued on next page*

*Continued*

| Reagent type (species) or resource | Designation | Source or reference | Identifiers | Additional information |
|---|---|---|---|---|
| Recombinant DNA reagent | pGL3-RARE-SV40-RFP (plasmid) | This paper | | Subcloned as described in Methods section |
| Recombinant DNA reagent | pGL3-RARE-SV40-d2EGFP (plasmid) | This paper | | Subcloned as described in Methods section |
| Recombinant DNA reagent | 12XTOPFLASH-d2EGFP (plasmid) | *Rios et al., 2010* | | |
| Recombinant DNA reagent | c*Raldh2* (plasmid) | *Diez del Corral et al., 2003* | | From K. Storey Template for probe synthesis |
| Recombinant DNA reagent | c*BAMBI* (plasmid) | *Casanova et al., 2012* | EST 603482731F1 | From A. Sanz-Ezquerro Template for probe synthesis |
| Recombinant DNA reagent | c*Cyp26A1* (plasmid) | *Wilson et al., 2007* | | From R. Wingate Template for probe synthesis |
| Recombinant DNA reagent | c*Hairy1* (plasmid) | *Jouve et al., 2000*; *Nitzan et al., 2016* | | From D. Henrique Template for probe synthesis |
| Recombinant DNA reagent | c*foxd3* (plasmid) | *Dottori et al., 2001* | EST 603374321F1 | From M. Cheung Template for probe synthesis |
| Recombinant DNA reagent | c*RARα* (plasmid) | *Diez del Corral et al., 2003* | | From K. Storey Template for probe synthesis |
| Recombinant DNA reagent | c*RARβ* (plasmid) | *Diez del Corral et al., 2003* | | From K. Storey Template for probe synthesis |
| Recombinant DNA reagent | c*RARγ* (plasmid) | A.Graham | | Template for probe synthesis |
| Recombinant DNA reagent | c*RXRα* (plasmid) | A.Graham | | Template for probe synthesis |
| Recombinant DNA reagent | c*RXRγ* (plasmid) | A.Graham | | Template for probe synthesis |
| Sequence-based reagent | q*Cyp1B1*_F | This paper | PCR primers | GTGTTGTGACTGCTGGGATG |
| Sequence-based reagent | q*Cyp1B1*_R | This paper | PCR primers | AGATTGACCAGTGAGCCAGG |
| Sequence-based reagent | q*sox9*_F | This paper | PCR primers | TCGAAGGAAACTGGCTGACC |
| Sequence-based reagent | q*sox9*_R | This paper | PCR primers | ATCAATGTGGGGAGGTTGGC |
| Sequence-based reagent | q*Rspo1*_F | This paper | PCR primers | AAACCACCGGTCTCTGTGTC |
| Sequence-based reagent | q*Rspo1*_R | This paper | PCR primers | AGCAGGAGGGAAGGAAGAAG |
| Sequence-based reagent | q*draxin*_F | This paper | PCR primers | TGTGCTGGATGTGGTTGTTT |
| Sequence-based reagent | q*draxin*_R | This paper | PCR primers | TGGTTTGCAGAGATGCTCAC |
| Sequence-based reagent | q*Grem1*_F | This paper | PCR primers | AGGCTGCTTTTGGAGAACAA |
| Sequence-based reagent | q*Grem1*_R | This paper | PCR primers | GAATGGGTTTTGGTTGATG |
| Sequence-based reagent | q*CRABP1*_F | This paper | PCR primers | ACCTGGAAGATGAGGAGCAG |
| Sequence-based reagent | q*CRABP1*_R | This paper | PCR primers | CACACGGTCACATACAACACC |
| Sequence-based reagent | q*Norrin(NDP)*_F | This paper | PCR primers | GCACTGTCCTAAAGCAGCCT |
| Sequence-based reagent | q*Norrin(NDP)*_R | This paper | PCR primers | TTCAGGCCCCGGGAGATATT |
| Sequence-based reagent | q*snai2*_F | This paper | PCR primers | TGAGATACGGGGAAAGACGC |
| Sequence-based reagent | q*snai2*_R | This paper | PCR primers | AGGCACTTGGAGGGGTAATG |
| Commercial assay or kit | KAPA2G Fast ReadyMix PCR | Sigma, Israel | Cat#KK5101 | |
| Chemical compound, drug | NBT | Roche | Cat#11383213001 | |
| Chemical compound, drug | BCIP | Sigma, Israel | Cat#B8503 | |
| Chemical compound, drug | Fast Red | Sigma, Israel | Cat#F4648 | |

*Continued on next page*

*Continued*

| Reagent type (species) or resource | Designation | Source or reference | Identifiers | Additional information |
|---|---|---|---|---|
| Software, algorithm | FIJI | *Schindelin et al., 2009* | https://imagej.net/software/fiji/ | |
| Software, algorithm | R and R-Studio | https://www.R-project.org; http://www.rstudio.com | RRID:SCR_000432 | R Version 4.0.3 |
| Software, algorithm | Adobe Photoshop and Indesign CS6 | Adobe | https://www.adobe.com/ | |
| Software, algorithm | Graphpad Prism | Graphpad https://www.graphpad.com | RRID:SCR_002798 | Version 7.0 |
| Other | BTX square wave electroporator | BTX, San Diego, CA, USA | Cat#45–0662 | |
| Other | DP73 cooled CCD digital camera | Olympus | https://www.olympus-global.com/ | |
| Other | BX51 microscope | Olympus | https://www.olympus-global.com/ | |
| Other | Nikon Eclipse 90i microscope | Nikon | https://www.nikon.com/ | |
| Other | D-Eclipse c1 confocal system | Nikon | https://www.nikon.com/ | |
| Other | DIG RNA mix | Roche | Cat#11277073910 | |
| Other | anti-digoxigenin-AP | Roche | Cat#11093274910 | |
| Other | Hoechst 33,258 stain | Sigma, Israel | Cat#14,530 | (125 ng/ml) |

## Embryos

Fertilized quail (*Coturnix coturnix Japonica*) eggs were obtained from commercial sources (Moshav Mata), kept at 15 °C and then incubated at 38 °C to desired stages. Embryos were staged by the number of somite pairs formed (somite stage – ss). All experiments were conducted at the flank level of the axis.

## Expression vectors and in ovo electroporation

The following expression vectors were used as controls: pCAGG, pCAGGS-EGFP (*Krispin et al., 2010b*), pCAG-mGFP (Addgene #14757), pCAGGS-RFP (*Ofek et al., 2021*). PCAGGS-RARα403 and PCAGGS-Cyp26A1 subcloned from PCAGGS-RARα403-IRES-eGFP and PCAGGS-Cyp26A1-IRES-eGFP, respectively, from S. Sockanathan (*Novitch et al., 2003*), were applied to inhibit RA activity. pCAB-cSmad6 was implemented to inhibit BMP activity (*Nitzan et al., 2016*). pCAG-VP16-RARα-IRES-eGFP was used to activate RA activity (*Novitch et al., 2003*).

To monitor RA activity, we used the pGL3-RARE-SV40-AP reporter [from J. Sen, *Gupta and Sen, 2015*]. Fluorescent versions of this reporter were prepared by replacing the AP tag with either RFP or with a destabilized version of GFP pGL3-RARE-SV40-RFP and pGL3-RARE-SV40-d2EGFP, respectively (Clontech). Wnt activity was measured with 12XTOPFLASH-d2EGFP (*Rios et al., 2010*).

Unilateral, bilateral and/or double electroporations to the NT were performed as detailed under Results. To this end, DNA (4 ug/ul) was mixed with Fast Green and microinjected into the lumen of the NT at the flank level of the axis. Five mm tungsten electrodes were placed on either side of the embryo. A square wave electroporator (BTX, San Diego, CA, USA) was used to deliver one pulse of current at 17–25 V for 6ms.

## Mechanical separation of the paraxial mesoderm from the NT

A unilateral slit between epithelial somites and NT comprising an approximate length of 8 segments was performed. A piece of aluminum foil was then inserted that extended beyond the dorsal border of the tissues to provide a continuous barrier despite embryo growth. In some cases, the dorsal NT was additionally opened to separate between both halves of the neuroepithelium. Following micro-surgery, embryos were reincubated for 48 hr prior to fixation.

## Immunohistochemistry

For immunostaining, embryos were fixed overnight at 4 °C with 4% formaldehyde (PFA) in phosphate-buffered saline (PBS) (pH 7.4), embedded in paraffin wax and serially sectioned at 8 μm.

Immunostaining was performed either on whole mount embryos or on paraffin sections, as previously described (*Burstyn-Cohen and Kalcheim, 2002*; *Kahane and Kalcheim, 2020*). Antibodies were diluted in PBS containing 5% fetal bovine serum (Biological Industries Israel, 04-007-1A) and 1% or 0.1% Triton X-100 (Sigma Aldrich Israel, X-100), respectively. Antibodies used were: rabbit anti-GFP (1:1000, Invitrogen, Thermo-Fisher Scientific, A-6455), mouse anti-GFP (1:100, Abcam, Ab38689), rabbit anti-RFP (1:1000, OriGene, AP09229PU-N), guinea pig anti-pSmad1/5/8 (1:300, a gift from E. Laufer), mouse anti-H3-pS10 (1:400, Abcam, Ab14955), rabbit anti-Laminin (1:100, Sigma Israel, L9393), rabbit anti-Sox9 (1:150, Millipore, AB5535), rabbit anti-SNAI2 (1:500; Cell Signaling Technologies, 9585), mouse anti-CD57 (anti-HNK1, 1:500, BD Biosciences, 559048), rabbit anti-BarHL1 (1:300, Sigma Israel, HPA004809), mouse anti-Pax7 (1:10, DHSB, pax7), mouse anti-Hb9 (1:200, DHSB, 81.5C10). Cell nuclei were visualized with 125 ng/ml Hoechst 33,258 (Sigma Israel, 14530) diluted in PBS.

## In situ Hybridization

For in situ hybridization (ISH), embryos were fixed with Fornoy (60% ethanol, 30% formaldehyde and 10% acetic acid) for 1 hr at room temperature, embedded in paraffin wax and serially sectioned at 10 µm. ISH was performed on sections. In brief, sections were treated with 1 µg/ml proteinase K, refixed in 4% PFA, then hybridized overnight at 65 °C with digoxigenin-labeled RNA probes (Roche, 11277073910). The probes were detected with AP coupled with anti-digoxigenin Fab fragments (Roche, 11093274910). AP reaction was developed with 4-Nitro blue tetrazolium chloride (NBT, Roche, 11383213001) and 5-bromo-4-chloro-3′-indolyphosphate p-toluidine salt (BCIP, Sigma Israel, B8503). In the *foxd3/Rspo1* ISH experiment, the AP reaction of *Rspo1* was developed with Fast Red (Sigma Israel, F4648) for 2 hr at room temperature. Importantly, all hybridizations, whether done on intact embryos at NC vs. RP stages, or in control vs. experimental embryos, were always developed for the same time per given probe and experiment. The RNA probes used are described in the Key Resources Table. Several probes were produced by PCR amplification (KAPA2G Fast ReadyMix PCR kit, Sigma Israel, KK5101). cDNA templates were synthesized by RNA precipitation followed by reverse transcription PCR. RNAs were produced from 20ss-E4 quail embryos. Tissue samples were homogenized with TriFast reagent, and RNA was separated with chloroform and isopropanol. The primers used are depicted in Key Resources Table.

## Data acquisition and statistical analysis

Measurement of the intensity of immunofluorescence and ISH signals: Intensities were measured using FIJI (*Schindelin et al., 2009*). In most cases, the RP was selected as the region of interest (ROI); mean intensity was measured and background intensity was then subtracted. Average intensity of all sections was calculated per embryo, and values were normalized with a control mean set as 1.

For measurement of phospho-Smad1/5/9 intensity, the ROI was selected as the dorsal domain of the NT from the dorsal midline to the ventral border of dI1 progenitors.

To monitor intensity of laminin immunostaining in the RP basal lamina, a segmented line of width four was drawn along the basal aspect of the RP, and mean intensity was measured within that area. Average intensity was calculated for each embryo and expressed relative to a control value set as 1.

To measure the relative position of cell nuclei across the apico-basal extent of the RP, the latter was divided into basal and apical halves, and Hoechst-stained nuclei were manually counted in both domains.

The distance of BarHL1[+] cells from the dorsal midline of the NT was measured for each cell manually using FIJI, and histograms were generated using R (https://www.R-project.org/) in Rstudio (http://www.rstudio.com/).

Images were photographed using a DP73 (Olympus) cooled CCD digital camera mounted on a BX51 microscope (Olympus) with Uplan FL-N 20 x/0.5 and 40 x/0.75 dry objectives (Olympus). Confocal sections encompassing their entire thickness were photographed using a Nikon Eclipse 90i microscope with a Plan Apo 40 x/1.3 or 100 x/1.4 objectives (Nikon) and a D-Eclipse c1 confocal system (Nikon) at 1.0 µm increments through the z-axis. Images were z-stacked with FIJI software.

For quantification, images of control and treated sections were photographed under the same conditions. For figure preparation, images were exported into Photoshop CS6 (Adobe). If necessary, the levels of brightness and contrast were adjusted to the entire image and images were cropped

without color correction adjustments or γ adjustments. Graphics were generated using Graphpad Prism 7.0, and figures were prepared using Photoshop and InDesign CS6.

Results were processed with Graphpad Prism seven and presented as scatter plots with mean ± SEM. Data were subjected to statistical analysis using either of the following tests, as described in the respective legends to Figures: Student's t-test, Welch's t-test, Mann-Whitney test, one-way ANOVA with post-hoc Tukey test, Kruskal-Wallis with post-hoc Dunn's test and Wilcoxon signed ranks test. All tests applied were two-tailed and a p-value ≤ 0.05 was considered significant.

## Acknowledgements

We are grateful to all the researchers whose names are mentioned along the text for generously providing reagents. We thank Avihu Klar and Joel Yisraeli for critical reading of the manuscript. This study was supported by grants from the Israel Science Foundation (ISF #209/18) and from the Ines Mandl Research Fund (IMRF) to CK.

## Additional information

### Funding

| Funder | Grant reference number | Author |
| --- | --- | --- |
| Israel Science Foundation | 209/18 | Chaya Kalcheim |
| Ines Mandl Research Fund | | Chaya Kalcheim |

The funders had no role in study design, data collection and interpretation, or the decision to submit the work for publication.

### Author contributions

Dina Rekler, Conceptualization, Data curation, Formal analysis, Investigation, Methodology, Validation, Visualization, Writing - original draft, Writing - review and editing; Chaya Kalcheim, Conceptualization, Funding acquisition, Investigation, Methodology, Project administration, Supervision, Writing - original draft, Writing - review and editing

### Author ORCIDs
Dina Rekler ⓘ http://orcid.org/0000-0001-7155-8712
Chaya Kalcheim ⓘ http://orcid.org/0000-0002-4612-9438

### Decision letter and Author response
Decision letter https://doi.org/10.7554/eLife.72723.sa1
Author response https://doi.org/10.7554/eLife.72723.sa2

## Additional files

### Supplementary files
• Transparent reporting form

### Data availability
All data generated or analysed during this study are included in the manuscript, figure supplements and source data files.

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
