## [Editor Report]

This manuscript examines how retinoic acid signaling controls the timing of neural crest production in avian embryos. The authors propose that local production of retinoic acid signaling activates the expression of BMP inhibitors in the dorsal neural tube. Disruption of BMP signaling results in the termination of neural crest migration and the establishment of the definite roof plate. In the absence of RA signaling cells in the dorsal neural tube continue to express neural crest markers well into the roof plate stage, and neural crest delamination is prolonged, but interestingly this does not prevent roof plate-mediated specification of dorsal interneurons.

---

## [Decision Letter]

**Decision letter after peer review:**

Thank you for submitting your article "Completion of neural crest cell production and emigration is regulated by retinoic acid-dependent inhibition of BMP signaling" for consideration by *eLife*. Your article has been reviewed by 2 peer reviewers, and the evaluation has been overseen by a Reviewing Editor and Didier Stainier as the Senior Editor. The following individual involved in review of your submission has agreed to reveal their identity: Laura Kerosuo (Reviewer #2).

Essential revisions:

The reviewers felt that in principle the finding that RA produced by the DNT is a key regulator of the transition of the dorsal neural tube to the definite roof plate is novel and of interest, however, they felt that showing just the expression pattern of RA associated genes and the one phenotype with prolonged NC features during RP stage caused by inhibition of RA signaling is not sufficient for publication in *eLife*; more details on the molecular mechanism are needed. In particular it was felt that experiments were needed to address whether RA is sufficient to turn the RP switch on prematurely if activated earlier in the DNT, and to clarify the reciprocal relationship and molecular mechanisms behind the crosstalk of RA and BMP inhibition. Moreover, since the early influence of RA produced by the somites (Martinez-Morales, 2011) seems to have an opposite effect than the RA shown here expressed in the DNT, it is important to provide mechanistic insights as to why that is. We would be happy to consider a revised manuscript that addresses the reviewers' concern on these three points as detailed below.

1) The authors must include additional experiments to examine the effects of early/ectopic activation of RA in the dorsal neural tube during premigratory stage to determine if it's sufficient to prematurely shut down neural crest production /delamination and switch to formation of RP.

2) Further examination of the proposed linkage between inhibition of BMP signaling and activation of RA is needed.

The authors should examine changes in expression of BMP and its inhibitors in the dorsal neural tube following up and down regulation of RA. Ideally this would be done quantitatively using qPCR. It might also be useful to include heatmaps generated from the existing RNaseq data comparing NC and RP showing expression changes of genes associated with RA, BMP and wnt signalling.

The experiment in figure 2 A-C needs an earlier time point to confirm that inhibition of RA indeed inhibits downregulation of pSMAD activity (and doesn't activate it, which could be another interpretation based on the current figure). If the author's model is correct an earlier time point should show the higher initial level of pSmad also in the control (which should be quantified to be comparable to the level seen in RAR403 treated embryos in Figure 2), becomes downregulated in the control by normal developmental time, but maintained in RARa403 and Cyp26A1 treated embryos.

The logic behind the interpretation of results in figure 5 is unclear. The fact that smad6 alone doesn't cause a phenotype is understandable given the timing when it's injected along the endogenous BMP inhibitors and extra Smad6 thus has no job to do in the embryo at that stage, (unless it causes a premature ending of NC emigration which we don't see with the current analysis time point.) So, the fact that co-expression of RA inhibitor together with a BMP inhibitor prevents the prolonged emigration caused by RARa403 does not prove RA is upstream of BMP but rather only confirms that the ectopic Smad6 is able to do its job as a reducer of NC stage also on RARa403-manipulated prolonged NC after its endogenous expression time window has closed. These experiments don't prove the reciprocal relationship or even a linkage between the two signalling pathways. Further experiments are needed to show the linkage – ideally biochemical experiments that would elucidate mechanism.

3) The role of wnt activation in response to changes in RA activation levels and timing should also be analyzed in more detail to understand the order of events. How is Wnt signaling impacted by premature activation of RA? In addition, an earlier time point is needed in sup Figure 2 to determine if activated wnt in the current figure is maintained from the NC stage of re-activated due to lack of RA. Currently wnt is addressed only in one supplementary figure and is not well integrated with the rest of the work.

4) In the current manuscript the mechanisms resulting in the opposing functions of RA starting or terminating neural crest production are only speculated upon. A revised manuscript should include experiments that shed light on how these distinct outcomes are manifested. Experiments that lend mechanistic insights into How RA inhibits RA signaling at RP stages are probably the key here. Are these outcomes of RA signaling not seen in response to early RA produced by the somites?

Other points:

1) Throughout the manuscript, the staging of the embryos and the axial levels analyzed in different experiments are loosely defined. Since the study focuses on timing, these variables are crucial and should be standardized and displayed in the figures/figure legends. Similarly please detail how all the variations of electroporation and injections were performed

2) The roof plate has been damaged in histological sections throughout the study, which prevents a clear assessment of the expression patterns described by the authors (e.g., Figure 1B, Figure S1I, Figure 3G).

3) The authors should demonstrate that the RARE construct is not active during the "Early NC" stages, which would allow for upregulation of BAMBI and the production of neural crest cells.

4) The illustration of the model in figure 9C should have no blue in the dorsal NT (to show the absence of RA).

5) In Figure 7 did the position of dl1 interneurons change because cells delaminated and migrated out? Is there a change in the number of cells that occupy the roof plate?

6) In Figure 5b' it looks like the control embryo also has GFP positive nuclei migrating from the tube in a row mostly on the right side but some on the left as well indicting emigration has not ceased in the controls either.

7) Supplementary Figure 1 is not convincing. Even though the expression is claimed to be ubiquitous, those images should not be shown because they might also be messy because of unspecific background staining.

8) The *Sox9* immunostaining in figure 3G looks backgroundy – it looks as though the mesenchyme and the epidermis are also positive, which don't express *Sox9* RNA according to 3E and F. The staining should be optimized.

9) Many of the references are papers from the authors' lab. When appropriate, please discuss the current findings in the broader context of others' work form others in the field.

*Reviewer #1 (Recommendations for the authors):*

– Throughout the manuscript, the staging of the embryos and the axial levels analyzed in different experiments are loosely defined. Since the study focuses on timing, these variables are crucial and should be standardized and displayed in the figures/figure legends.

– The timing of expression of the BMP inhibitors is a crucial part of the model proposed by the authors. If these genes are downstream of NT-derived RA, they should only be transcribed during roof plate stages.

– The roof plate has been damaged in histological sections throughout the study, which prevents a clear assessment of the expression patterns described by the authors (e.g., Figure 1B, Figure S1I, Figure 3G).

– Figure 7 – did the position of dl1 interneurons change because cells delaminated and migrated out? Is there a change in the number of cells that occupy the roof plate?

– The authors should demonstrate that the RARE construct is not active during the "Early NC" stages, which would allow for upregulation of BAMBI and the production of neural crest cells.

– The illustration of the model in figure 9C should have no blue in the dorsal NT (to show the absence of RA).

---

## [Author Response]

Essential revisions:The reviewers felt that in principle the finding that RA produced by the DNT is a key regulator of the transition of the dorsal neural tube to the definite roof plate is novel and of interest, however, they felt that showing just the expression pattern of RA associated genes and the one phenotype with prolonged NC features during RP stage caused by inhibition of RA signaling is not sufficient for publication in eLife; more details on the molecular mechanism are needed. In particular it was felt that experiments were needed to address whether RA is sufficient to turn the RP switch on prematurely if activated earlier in the DNT, and to clarify the reciprocal relationship and molecular mechanisms behind the crosstalk of RA and BMP inhibition. Moreover, since the early influence of RA produced by the somites (Martinez-Morales, 2011) seems to have an opposite effect than the RA shown here expressed in the DNT, it is important to provide mechanistic insights as to why that is. We would be happy to consider a revised manuscript that addresses the reviewers' concern on these three points as detailed below.

We thank the reviewers for their interest in our findings. In the original manuscript we reported that neural tubederived RA acts 1) to upregulate expression of BMP inhibitors, 2) to inhibit NC-specific genes, 3) to abrogate BMP signaling and 4) consequently to end the period of NC production/emigration. 5) In addition, that RA is not necessary for RP or dorsal interneuron (dI1) formation, but it is required for proper lineage segregation of NC/RP and separation of all the above cell types both in time and space. 6) that somite-derived RA is dispensable for expression of RAdependent traits in RP.

We have now addressed to the best of our ability the three main issues raised during the review process.

First, we find that premature gain of RA function is not sufficient to promote an earlier termination of BMP signaling, NC gene expression or NC emigration, or to cause a premature appearance of RP genes, meaning that dorsal NTderived RA is necessary but not sufficient for these processes. Further elaboration of this issue is detailed below. Second, we further clarify the reciprocal relationship behind the cross talk between RA and BMP. We find that both pathways stand in a mutual repressive interaction that ensures the correct temporal progression of events in the dorsal NT.

Third, we clarify that the effects of RA during NC and RP development are not simply of an opposite binary nature, they are much more complex and context dependent. In this article, we show for the first time the molecular nature of the differential activities and propose several mechanisms to explain them, yet believe that demonstration of direct mechanistic interactions, while being the next step, is beyond the scope of this study. Below, we address each comment, as follows:

1) The authors must include additional experiments to examine the effects of early/ectopic activation of RA in the dorsal neural tube during premigratory stage to determine if it's sufficient to prematurely shut down neural crest production /delamination and switch to formation of RP.

As requested, we have now examined the effect of gain of RA function on the above processes. To note is that RA is active in the dorsal NT at the NC stage, where BMP is also active, premigratory cells express specific NC genes and NC EMT takes place. Hence, these NC-related events are fully compatible with the presence of RA activity in the NT. So, the gain of function experiments requested only add extra-RA signaling to the system.

We performed careful serial section analyses using VP16-RAR-α to activate RA signaling (corroborated to work in our system by destabilized RARE-GFP) at very precise stages. As shown in new Figure 5, Supp.2, Figure 5. Supp.3, and Figure 6, Supp.2, we found no significant differences in the timing of decline of BMP signaling, *Sox9* or *foxd3*

expressions or on NC EMT, showing that excess RA activity does not cause a premature cessation of NC-specific traits. Reciprocally, we examined whether gain of RA function is able to advance the onset of expression of selected RP traits and we also found no effect on the first appearance of *BAMBI* or *Rspo1.* The latter data confirm additional results suggesting that the end of NC and advent of RP are mostly regulated by independent factors.

Together, our results show that RA is necessary but not sufficient for the end of NC production. Thus it is not the sole activity of RA in the NT that causes the end of NC EMT but rather a time-specific interplay between RA signaling in the context of a gene network that changes between NC and nascent RP stages.

2) Further examination of the proposed linkage between inhibition of BMP signaling and activation of RA is needed.The authors should examine changes in expression of BMP and its inhibitors in the dorsal neural tube following up and down regulation of RA. Ideally this would be done quantitatively using qPCR.

The loss of function experiments were already presented in the original manuscript and the gain of function data were now performed and described in the previous item (see above). The referee requests we address changes in BMP expression- to note is that all BMP proteins are produced in the dNT across both stages albeit at changing levels (see Figure 1 Supp.1), while the responsiveness to BMP signaling is the process that is downregulated in the transition to definitive RP.

As to quantification by qPCR- the referee may notice that most genes examined are not restricted solely to the dorsal NT/RP domains. Since it is technically not accurate to isolate only the regions of interest for qPCR analysis, collecting entire NTs following unilateral or bilateral electroporations for qPCR would be highly imprecise. In situ hybridization and immunohistochemistry provide precise tools to assess the spatial localization of the transcripts/proteins of interest. To note is that in all cases examined, development of the color reactions was for the same length of time for control and experimental cases and photography was performed under identical conditions. Furthermore, in most cases, effects between treatments were dramatic, readily apparent at a qualitative level and easily quantifiable from ISH or fluorescent images.

It might also be useful to include heatmaps generated from the existing RNaseq data comparing NC and RP showing expression changes of genes associated with RA, BMP and wnt signalling.

A new figure, supplementary 1 to Figure 1, presents data of our transcriptome analysis focusing on differentially expressed genes between NC and RP stages regarding RA, BMP and Wnt pathways, as requested.

The experiment in figure 2 A-C needs an earlier time point to confirm that inhibition of RA indeed inhibits downregulation of pSMAD activity (and doesn't activate it, which could be another interpretation based on the current figure). If the author's model is correct an earlier time point should show the higher initial level of pSmad also in the control (which should be quantified to be comparable to the level seen in RAR403 treated embryos in Figure 2), becomes downregulated in the control by normal developmental time, but maintained in RARa403 and Cyp26A1 treated embryos.

As requested, an earlier time point was now performed to confirm that pSMAD expression (e.g; BMP activity) is present at the NC stage. This is depicted now in revised Figure 2, panels A-D’. Because at the RP stage both RP and interneurons continue expressing pSMAD when compared to the NC stage where a homogeneous expression is detected in the dorsal NT, we inferred that a comparative quantification of both stages would not be appropriate. Therefore, quantification in Figure 2E compares control and treated conditions at the RP stage exclusively. Having said that, the intensities at both stages are very similar, as shown in panels A,C, and D which were photographed at exactly the same conditions. Hence, inhibition of RA signaling prolongs BMP activity into the RP stage by preventing its downregulation.

The logic behind the interpretation of results in figure 5 is unclear. The fact that smad6 alone doesn't cause a phenotype is understandable given the timing when it's injected along the endogenous BMP inhibitors and extra Smad6 thus has no job to do in the embryo at that stage, (unless it causes a premature ending of NC emigration which we don't see with the current analysis time point.) So, the fact that co-expression of RA inhibitor together with a BMP inhibitor prevents the prolonged emigration caused by RARa403 does not prove RA is upstream of BMP but rather only confirms that the ectopic Smad6 is able to do its job as a reducer of NC stage also on RARa403-manipulated prolonged NC after its endogenous expression time window has closed. These experiments don't prove the reciprocal relationship or even a linkage between the two signalling pathways. Further experiments are needed to show the linkage – ideally biochemical experiments that would elucidate mechanism.

In the first part of the manuscript, we reported that both RARa403 and Cyp26a1 inhibit the onset of expression of BMP inhibitors exclusively transcribed in RP, extend BMP signaling into the RP stage where it is not normally active, and extend NC EMT beyond its normal time period.

Despite these findings linking late RA to BMP signaling pathways, we were troubled by the possibility that the extended EMT induced by inhibiting RA signaling could be mediated by factors other than BMP. If so, we reasoned that inhibiting BMP by Smad6 would not prevent RARa403 from stimulating the “late” NC emigration. This was not the case, as Smad6, which inhibits BMP at early stages (Nitzan et al., 2016), also prevented the extended/late emigration elicited by lack of RA activity. In our view, this finding provides a clear link between both pathways. We have now modified the text and hope this clarifies this concern.

3)The role of wnt activation in response to changes in RA activation levels and timing should also be analyzed in more detail to understand the order of events. How is Wnt signaling impacted by premature activation of RA? In addition, an earlier time point is needed in sup Figure 2 to determine if activated wnt in the current figure is maintained from the NC stage of re-activated due to lack of RA. Currently wnt is addressed only in one supplementary figure and is not well integrated with the rest of the work.

Due to technical limitations of combining VP-16 and the destabilized Topflash reporter for Wnt signaling, the effect of gain of RA function on Wnt activity could not be examined. As requested, an earlier time point showing significant Wnt activity in the dorsal NT at the NC stage was added. Since most activity at the RP stage in RA-repressed conditions was detected in the RP domain (Revised Figure 2 Supplem. 2), we could also quantitatively compare NC and RP stages. Clearly, under normal conditions, Wnt activity in RP is greatly reduced when compared to premigratory NC, and this reduction is prevented to a significant extent by inhibiting RA signaling (panel D). Hence, similar to the situation regarding BMP activity, inhibition of RA signaling prolongs Wnt activity into the RP stage by preventing its downregulation rather than by de novo activating it. The similar effects of RA deprivation on both BMP and Wnt activities is expected, as at the NC stage, we previously showed that the order of events is such that Wnt acts downstream of BMP in the control of NC EMT (Burstyn-Cohen et al., 2004).

4) In the current manuscript the mechanisms resulting in the opposing functions of RA starting or terminating neural crest production are only speculated upon. A revised manuscript should include experiments that shed light on how these distinct outcomes are manifested. Experiments that lend mechanistic insights into How RA inhibits RA signaling at RP stages are probably the key here.

We thank the referees for this comment. We think that a key question is understanding how RA signaling is differentially interpreted over time given its multistage activity in dorsal NT development.

This notion is based on the following findings: Years ago, we uncovered that the balance between activities of BMP/Wnt and noggin in the dorsal NT triggers the onset of NC EMT (Sela-Donenfeld and Kalcheim, Burstyn- Cohen et al.,). Martinez-Morales et al., strengthened our findings by reporting that a balance between somitic RA and FGF works on the reported BMP/Wnt modules to initiate the process. This group found that at gastrulation stages, RA is required for NC specification, as revealed by analysis of VAD quail embryos. Next, during somite formation, somitic RA is necessary for the onset of emigration of specified NC progenitors but at advanced somite stages it is dispensable for the subsequent maintenance of cell emigration. Presently, we find that RP-derived RA ends NC production. Together, this highlights a dynamic behavior of RA at 4 sequential stages of NC ontogeny. Clearly enough, the two first effects are mediated by an influence of RA dorso-ventral patterning of the early NT, as distribution of ventral NT markers was strongly affected. In our case, RA from the nascent RP has no such effects suggesting that RP-derived RA acts at a post-patterning phase to specifically affect the dorsal NT.

All things considered, we think that the problem is not simply a binary question of “opposing functions of RA signaling in starting or terminating NC production”. Instead, it is the understanding of a differential interpretation to the same morphogen by progenitor cells with changing states and at sequential stages.

To the referee’s request, we begun addressing the question of how does RA inhibit BMP signaling close to the RP stage. To this end, it was first necessary to examine the temporal regulation of *Raldh2* expression that is restricted to the RP stage, and is therefore a prerequisite for the late activity of RA. Whereas repressing RA activity extends the NC phase including the continuous transcription of Foxd3, *Sox9* and Snail2 (Figure 3), we now found that preventing the normal downregulation of these factors by extending their activity into the RP stage, represses the onset of *Raldh2* transcription in the nascent RP (new Figure 9). We interpret these results to mean that as long as NC genes are active in the dorsal NT (NC stage), local Raldh2 and consequent RA synthesis in the NT does not take place, so Raldh2 in RP is repressed by NC-specific traits.

The significance of these data is twofold: first, they explain how the onset of Raldh2 production is restricted to the RP stage. Second, since we also report the reciprocal result, e.g; that RA represses NC genes (Figure 3), we conclude that a cross-repressive interaction exists between NC and RP-specific genes downstream of RA, being an emerging temporal property of the network. These data further indicate that the changing roles of RA throughout development of the dorsal neural primordium, largely depend on a different interpretation of the signal mediated by changing and mutually repressive codes.

We have now presented these data in new Figure 9. To clarify our thoughts further, we now provide a working model summarizing the effects of RA in NC to RP transition (Figure 10B).

Our article uncovers for the first time and thoroughly documents, a role of local RA activity on the end of NC production and ensuing RP architecture. We believe that a comprehensive elucidation of the molecular mechanism responsible for inhibition of BMP signaling by local RA is the next obligatory step. We show in this study the selective activation of BMP inhibitors by endogenous RA and previously found that one of them, Hes/hairy, indeed inhibits BMP signaling and NC EMT (Nitzan et al., 2016). Therefore we propose (not only speculate) that upregulation of BMP inhibitors by RA is a possible mechanism. However, we also predict that this is not the only one, and a deeper understanding of this problem is beyond the scope of the present study.

Additional possibilities that fit with our data were now discussed: RA expression in somites vs. RP can be regulated by different enhancers and thus have distinct functions. For example, a specific enhancer driving expression of Raldh2 was found to be activated only at the definitive RP stage (Castillo et al., 2010). This enhancer contains Tcf binding sites and thus may be activated by Wnt signaling. In turn, as we show, RP-derived Raldh2 and resulting RA could negatively feed-back on Wnt signaling in the formed RP either directly or through BMP acting upstream of Wnt (now summaized in Figure 10B).

Another possible scenario is that RA represses BMP signaling by inactivating Smad proteins via ubiquitination, as shown to be the case in selected cell lines (Sheng et al., 2010). These possibilities were now discussed and await to be explored systematically.

Are these outcomes of RA signaling not seen in response to early RA produced by the somites?

We have now examined BMP and Wnt activities in the dorsal NT at early stages, when RA is only provided from the somites, and found that both factors are clearly active at this stage (Revised Figure 2 and Revised Figure 2 Supp.2, and see also Nitzan 2016). Hence, somitic RA is compatible at the NC stage with BMP/Wnt activities in the NT. In contrast, no significant BMP or Wnt signals are detected at the RP stage when RA is locally produced (our original findings). Furthermore, separation of somites from the NT to prevent access of somitic RA to the NT (and of other factors as well), has no effect on expression of *BAMBI*, a direct readout of RA activity in the RP (Figure 8). In fact, many studies demonstrated that emigration of NC cells takes place even in the absence of somites when NTs are explanted on plain fibronectin. The role of the somites is, therefore, to impose a timer on initiation of the process (Sela-Donenfeld and Kalcheim, 1998), and this was corroborated by Diez del Corral et al., who found that only the onset, but not the maintenance of sequential NC EMT is dependent on somitic RA. Thus, for these reasons and those elaborated in the item above, we can say that the end of NC EMT and additional phenotypes here described are not seen in response to mesodermal RA.

Other points:1) Throughout the manuscript, the staging of the embryos and the axial levels analyzed in different experiments are loosely defined. Since the study focuses on timing, these variables are crucial and should be standardized and displayed in the figures/figure legends. Similarly please detail how all the variations of electroporation and injections were performed

We agree that timing is of the essence here. The information requested was originally provided in the corresponding legends. We re-examined carefully the text and added missing information wherever relevant. We opted for providing full details in each legend in order not to interfere with the flow of the text.

2) The roof plate has been damaged in histological sections throughout the study, which prevents a clear assessment of the expression patterns described by the authors (e.g., Figure 1B, Figure S1I, Figure 3G).

Figure 1B and Figure 3G were corrected. The NC panels in Figure 1 S2 were deleted.

3) The authors should demonstrate that the RARE construct is not active during the "Early NC" stages, which would allow for upregulation of BAMBI and the production of neural crest cells.

As shown in the original version of the manuscript (Figure 1,N,N’, O,O’), the RARE construct is active both at NC and RP stages. This is consistent with data by others (Rossant et al., 1991, Olivera- Martinez and Storey, 2007) which also found RA activity in the early NT of mice and chick, respectively. The difference is that the early NT responds to RA but does not synthesize it, whereas the late NT both produces and also responds to the factor. Hence, selective expression of *BAMBI* at the RP stage associated with the completion of NC production cannot be solely explained by activity of lack of local activity of RA. Rather, by a differential interpretation of the signal by premigratory NC cells vs. nascent RP cells. As discussed, this may be accounted for by changes in efficiency of local RA activity, in the changing transcriptional profile of the cells, etc.

4) The illustration of the model in figure 9C should have no blue in the dorsal NT (to show the absence of RA).

The blue color represents Raldh2 expression. The reagents we used selectively repress RA activity, not Raldh2. We have now clarified this in the scheme. “RA synthesis (Raldh2)” was changed to “Raldh2 expression” (now Figure 10A).

5) In Figure 7 did the position of dl1 interneurons change because cells delaminated and migrated out? Is there a change in the number of cells that occupy the roof plate?

Most probably, dI1 interneurons invaded the roof plate because its cellular architecture is altered in the absence of RA signaling (see Figure 4) altogether compromising the integrity of this domain (e.g, lack of segregation between NC and RP fates , Figure 6), thus enabling mislocalization of interneurons otherwise located immediately ventral to the RP. As discussed originally, this may result from a change in the D-V gradient of BMP and synthesis of Slit1, both factors involved in axonal/cellular guidance, which are altered under experimental conditions and may affect cellular integrity and interneuron localization. Having said that, we cannot rule out the possibility raised by the referee that continuous NC cell emigration may also act as a driving force for more ventral cell types to undergo a dorsal shift, and this was now discussed.

To address the referee’s question on RP size, we now measured the area of the Rspo1-positive roof plate (which is a faithful representation of cell number, see Kahane and Kalcheim, 2020), and found no significant difference in its size between control and treated embryos. Results were added to the text (control: 1672 ± 121.3 µm^2^ [mean ± SEM, n = 8]; compared to RARa403: 1790 ± 146.1 µm^2^ [mean ± SEM, n=8], p = 0.544 [non-significant], student's t-test).

6) In Figure 5b' it looks like the control embryo also has GFP positive nuclei migrating from the tube in a row mostly on the right side but some on the left as well indicting emigration has not ceased in the controls either.

A single fluorescent cell is seen in the right side of the image. The others represent weak autofluorescence of erythrocytes. We aimed at performing the second electroporation at a post-EMT stage to monitor only the late emigration occurring at the RP stage. Occasionally, because analysis comprises a certain length of the axis, one observes at more caudal levels very few delaminating cells in controls. The referee may look at panels D and I showing the low values of cell emigration in controls; the enhancement observed upon repression of RA signaling is, however, dramatic.

7) Supplementary Figure 1 is not convincing. Even though the expression is claimed to be ubiquitous, those images should not be shown because they might also be messy because of unspecific background staining.

We agree with the referee that the expression patterns of RA receptors, as revealed primarily at the NC stage, may not be convincing. In this regard, we have looked at other studies and noticed, that RARs and RXRs also display a rather even distribution in the neural tube of early avian embryos (Diez del Corral et al., 2003). To meet the referee’s request, we decided to omit the images of the NC stage while acknowledging the reference by the Storey team. If acceptable, we would like to present the images depicting the RP stage where patterns look clearer and show the presence of RAR-α and RAR-β and RXR-γ (but not of RAR-γ or RXR-α) transcripts in the RP itself (Figure 1 Supp.2).

8) The Sox9 immunostaining in figure 3G looks backgroundy – it looks as though the mesenchyme and the epidermis are also positive, which don't express Sox9 RNA according to 3E and F. The staining should be optimized.

We removed the Hoechst staining that was in the background of these images and we believe this greatly improved them. In particular, the clarity of the region of interest, the RP. To note is that *Sox9* signal in mesenchyme is not background staining as *Sox9* is a key gene for chondrogenesis and begins to be expressed in ventral somite and then sclerotome, as also seen in 3E and F. The epidermal signal likely stems from superficial background of the wholemount immunostaining, it cannot always be avoided and does not interfere with the message.

9) Many of the references are papers from the authors' lab. When appropriate, please discuss the current findings in the broader context of others' work form others in the field.

As requested, several additional references were added to the Introduction and Discussion sections.

Reviewer #1 (Recommendations for the authors):– Throughout the manuscript, the staging of the embryos and the axial levels analyzed in different experiments are loosely defined. Since the study focuses on timing, these variables are crucial and should be standardized and displayed in the figures/figure legends.

We agree that timing is of the essence here. The information requested was originally provided in the corresponding legends. We re-examined carefully the text and added missing information wherever relevant. We opted for providing full details in each legend in order not to interfere with the flow of the text.

– The timing of expression of the BMP inhibitors is a crucial part of the model proposed by the authors. If these genes are downstream of NT-derived RA, they should only be transcribed during roof plate stages.

Indeed, they are only transcribed at RP stages and not in premigratory NC. This was thoroughly documented in a previous article from our lab. (Ofek et al., 2021) and now in Figure 1 Supplem.1 depicting heatmaps of relevant genes, including BMP inhibitors.

– The roof plate has been damaged in histological sections throughout the study, which prevents a clear assessment of the expression patterns described by the authors (e.g., Figure 1B, Figure S1I, Figure 3G).

Figure 1B and Figure 3G were corrected. Panels depicting NC stages in Figure 1 Supp.2 were deleted.

– Figure 7 – did the position of dl1 interneurons change because cells delaminated and migrated out? Is there a change in the number of cells that occupy the roof plate?

Most probably, dI1 interneurons invaded the roof plate because its cellular architecture is altered in the absence of RA signaling (see Figure 4) altogether compromising the integrity of this domain (e.g, lack of segregation between NC and RP fates , Figure 6), thus enabling mislocalization of interneurons otherwise located immediately ventral to the RP. As discussed originally, this may result from a change in the D-V gradient of BMP and synthesis of Slit1, both factors involved in axonal/cellular guidance, which are altered under experimental conditions and may affect cellular integrity and interneuron localization. Having said that, we cannot rule out the possibility raised by the referee that continuous NC cell emigration may also act as a driving force for more ventral cell types to undergo a dorsal shift, and this was now discussed.

To address the referee’s question on RP size, we now measured the area of the Rspo1-positive roof plate (which is a faithful representation of cell number, see Kahane and Kalcheim, 2020), and found no significant difference in its size between control and treated embryos. Results were added to the text (control: 1672 ± 121.3 µm^2^ [mean ± SEM, n = 8]; compared to RARa403: 1790 ± 146.1 µm^2^ [mean ± SEM, n=8], p = 0.544 [non-significant], student's t-test).

– The authors should demonstrate that the RARE construct is not active during the "Early NC" stages, which would allow for upregulation of BAMBI and the production of neural crest cells.

As shown in the original version of the manuscript (Figure 1,N,N’, O,O’), the RARE construct is active both at NC and RP stages. This is consistent with data by others (Rossant et al., 1991, Olivera- Martinez and Storey, 2007) which also found RA activity in the early NT of mice and chick, respectively. The difference is that the early NT responds to RA but does not synthesize it, whereas the late NT both produces and also responds to the factor. Hence, selective expression of *Bambi* at the RP stage associated with the completion of NC production cannot be solely explained by activity of lack of local activity of RA. Rather, by a differential interpretation of the signal by premigratory NC cells vs. nascent RP cells. As discussed in the revised manuscript, this may be accounted for by changes in local RA levels, in the changing transcriptional profile of the cells, etc.

– The illustration of the model in figure 9C should have no blue in the dorsal NT (to show the absence of RA).

The blue color represents Raldh2 expression, which it is not affected by our treatments. The reagents we used selectively repress RA activity. We have now clarified this in the scheme. “RA synthesis (Raldh2)” was changed to “Raldh2 expression” (now Figure 10A).